# Beyond Alternating Updates for Matrix Factorization with Inertial Bregman Proximal Gradient Algorithms

**Mahesh Chandra Mukkamala**
Mathematical Optimization Group
Saarland University, Germany
mukkamala@math.uni-sb.de

**Peter Ochs**
Mathematical Optimization Group
Saarland University, Germany
ochs@math.uni-sb.de

## Abstract

Matrix Factorization is a popular non-convex optimization problem, for which alternating minimization schemes are mostly used. They usually suffer from the major drawback that the solution is biased towards one of the optimization variables. A remedy is non-alternating schemes. However, due to a lack of Lipschitz continuity of the gradient in matrix factorization problems, convergence cannot be guaranteed. A recently developed approach relies on the concept of Bregman distances, which generalizes the standard Euclidean distance. We exploit this theory by proposing a novel Bregman distance for matrix factorization problems, which, at the same time, allows for simple/closed form update steps. Therefore, for non-alternating schemes, such as the recently introduced Bregman Proximal Gradient (BPG) method and an inertial variant Convex–Concave Inertial BPG (CoCaIn BPG), convergence of the whole sequence to a stationary point is proved for Matrix Factorization. In several experiments, we observe a superior performance of our non-alternating schemes in terms of speed and objective value at the limit point.

## 1 Introduction

Matrix factorization has numerous applications in Machine Learning [43, 57], Computer Vision [17, 58, 62, 28], Bio-informatics [56, 12] and many others. Given a matrix $\mathbf{A} \in \mathbb{R}^{M \times N}$, one is interested in the factors $\mathbf{U} \in \mathbb{R}^{M \times K}$ and $\mathbf{Z} \in \mathbb{R}^{K \times N}$ such that $\mathbf{A} \approx \mathbf{U}\mathbf{Z}$ holds. This is usually cast into the following non-convex optimization problem

$$\min_{\mathbf{U} \in \mathcal{U}, \mathbf{Z} \in \mathcal{Z}} \left\{ \Psi \equiv \frac{1}{2} \|\mathbf{A} - \mathbf{U}\mathbf{Z}\|_F^2 + \mathcal{R}_1(\mathbf{U}) + \mathcal{R}_2(\mathbf{Z}) \right\}, \tag{1.1}$$

where $\mathcal{U}, \mathcal{Z}$ are constraint sets and $\mathcal{R}_1, \mathcal{R}_2$ are regularization terms. The most frequently used techniques for solving matrix factorization problems involve alternating updates (Gauss–Seidel type methods [26]) like PALM [8], iPALM [53], BCD [63], BC-VMFB [18], HALS [19] and many others. A common disadvantage of these schemes is their bias towards one of the optimization variables. Such alternating schemes involve fixing a subset of variables to do the updates. In order to guarantee convergence to a stationary point, alternating schemes require the first term in (1.1) to have a Lipschitz continuous gradient only with respect to each subset of variables. However, in general Lipschitz continuity of the gradient fails to hold for all variables. The same problem appears in various practical applications such as Quadratic Inverse Problems, Poisson Linear Inverse Problems, Cubic Regularized Non-convex Quadratic Problems and Robust Denoising Problems with Non-convex Total Variation Regularization [46, 9, 4]. They belong to the following broad class of non-convex additive composite minimization problems

$$(\mathcal{P}) \qquad \inf \left\{ \Psi \equiv f(x) + g(x) : x \in \overline{C} \right\}, \tag{1.2}$$

where $f$ is potentially a non-convex extended real valued function, $g$ is a smooth (possibly non-convex) function and $\overline{C}$ is a nonempty, closed, convex set in $\mathbb{R}^d$. In order to use non-alternating schemes for (1.1), the gradient Lipschitz continuity must be generalized. Such a generalization was initially proposed by [6] and popularized by [4] in convex setting and for non-convex problems in [9]. They are based on a generalized proximity measure known as Bregman distance and have recently led to new algorithms to solve (1.2): Bregman Proximal Gradient (BPG) method [9] and its inertial variant Convex–Concave Inertial BPG (CoCaIn BPG) [46].

BPG generalizes the proximal gradient method from Euclidean distances to Bregman distances as proximity measures. Its convergence theory relies on the generalized Lipschitz assumption, discussed above, called $L$-smad property [9]. It involves an upper bound and a lower bound, where the upper bound involves a convex majorant to control the step-size of BPG. However, the significance of lower bounds for BPG was not clear. In non-convex optimization literature, the lower bounds which involve concave minorants were largely ignored. Recently, extending on [61, 50], CoCaIn BPG changed this trend by justifying the usage of lower bounds to incorporate inertia for faster convergence [46]. Moreover, the generated inertia is adaptive, in the sense that it changes according to the function behavior, i.e., CoCaIn BPG does not use an inertial parameter depending on the iteration counter unlike Nesterov Accelerated Gradient (NAG) method [47] (also FISTA [5]) in the convex setting.

In this paper we ask the question: *"Can we apply BPG and CoCaIn BPG efficiently for Matrix Factorization problems?"*. This question is significant, since convergence of the Bregman minimization variants BPG and CoCaIn BPG relies on the $L$-smad property, which is non-trivial to verify and an open problem for Matrix Factorization. Another crucial issue is the efficient computability of the algorithm's update steps, which is particularly hard due to the coupling between two subsets of variables. We successfully solve these challenges.

**Contributions.** We make recently introduced powerful Bregman minimization based algorithms BPG [9] and CoCaIn BPG [46] and the corresponding convergence results applicable to the matrix factorization problems. Experiments show a significant advantage of BPG and CoCaIn BPG which are non-alternating by construction, compared to popular alternating minimization schemes in particular PALM [8] and iPALM [53]. The proposed algorithms require the following non-trivial contributions:

- We propose a novel Bregman distance for Matrix Factorization with the following auxiliary function (called kernel generating distance) with certain $c_1, c_2 > 0$:

$$h(\mathbf{U}, \mathbf{Z}) = c_1 \left( \frac{\|\mathbf{U}\|_F^2 + \|\mathbf{Z}\|_F^2}{2} \right)^2 + c_2 \left( \frac{\|\mathbf{U}\|_F^2 + \|\mathbf{Z}\|_F^2}{2} \right) .$$

  The generated Bregman distance embeds the crucial coupling between the variables $\mathbf{U}$ and $\mathbf{Z}$. We prove the L-smad property with such a kernel generating distance and infer convergence of BPG and CoCaIn BPG to a stationary point.

- We compute the analytic solution for subproblems of the proposed variants of BPG, for which the usual analytic solutions based on Euclidean distances cannot be used.

**Simple Illustration of BPG for Matrix Factorization.** Consider the following simple matrix factorization optimization problem, where we set $\mathcal{R}_1 := 0$ and $\mathcal{R}_2 := 0$ in (1.1)

$$\min_{\mathbf{U} \in \mathbb{R}^{M \times K}, \mathbf{Z} \in \mathbb{R}^{K \times N}} \left\{ \Psi(\mathbf{U}, \mathbf{Z}) = \frac{1}{2} \|\mathbf{A} - \mathbf{U}\mathbf{Z}\|_F^2 \right\} . \tag{1.3}$$

For this problem, the update steps of **Bregman Proximal Gradient for Matrix Factorization (BPG-MF)** given in Section 2.1 (also see Section 2.4) with a chosen $\lambda \in (0, 1)$ are the following:

In each iteration, compute $t_k = 3(\|\mathbf{U}^\mathbf{k}\|_F^2 + \|\mathbf{Z}^\mathbf{k}\|_F^2) + \|\mathbf{A}\|_F$ and perform the intermediary gradient descent steps (non-alternating) for $\mathbf{U}$ and $\mathbf{Z}$ independently with step-size $\frac{\lambda}{t_k}$:

$$\mathbf{P}^\mathbf{k} = \mathbf{U}^\mathbf{k} - \frac{\lambda}{t_k} \left[ (\mathbf{U}^\mathbf{k}\mathbf{Z}^\mathbf{k} - \mathbf{A})(\mathbf{Z}^\mathbf{k})^T \right] , \quad \mathbf{Q}^\mathbf{k} = \mathbf{Z}^k - \frac{\lambda}{t_k} \left[ (\mathbf{U}^\mathbf{k})^T(\mathbf{U}^\mathbf{k}\mathbf{Z}^\mathbf{k} - \mathbf{A}) \right] .$$

Then, the additional scaling steps $\mathbf{U}^{\mathbf{k+1}} = rt_k\mathbf{P}^\mathbf{k}$ and $\mathbf{Z}^{\mathbf{k+1}} = rt_k\mathbf{Q}^\mathbf{k}$ are required, where the scaling factor $r \geq 0$ satisfies a cubic equation: $3t_k^2 \left( \|\mathbf{P}^\mathbf{k}\|_F^2 + \|\mathbf{Q}^\mathbf{k}\|_F^2 \right) r^3 + \|\mathbf{A}\|_F r - 1 = 0$.

## 1.1 Related Work

**Alternating Minimization** is the go-to strategy for matrix factorization problems due to coupling between two subsets of variables [24, 1, 64]. In the context of non-convex and non-smooth optimization, recently PALM [8] was proposed and convergence to stationary point was proved. An inertial variant, iPALM was proposed in [53]. However, such methods require a subset of variables to be fixed. We remove such a restriction here and take the contrary view by proposing non-alternating schemes based on a powerful Bregman proximal minimization framework, which we review below.

**Bregman Proximal Minimization** extends upon the standard proximal minimization, where Bregman distances are used as proximity measures. Based on initial works in [6, 4, 9], related inertial variants were proposed in [46, 67]. Related line-search methods were proposed in [52] based on [10, 11]. More related works in convex optimization include [49, 40, 42]. Recently, the symmetric non-negative matrix factorization problem was solved with a non-alternating Bregman proximal minimization scheme [21] with the following kernel generating distance

$$h(\mathbf{U}) = \frac{\|\mathbf{U}\|_F^4}{4} + \frac{\|\mathbf{U}\|_F^2}{2} .$$

However for the following applications, such a $h$ is not suitable, unlike our Bregman distance.

**Non-negative Matrix Factorization (NMF)** is a variant of the matrix factorization problem which requires the factors to have non-negative entries [25, 37]. Some applications are hyperspectral unmixing, clustering and others [24, 22]. The non-negativity constraints pose new challenges [37] and only convergence to a stationary point [24, 31] is guaranteed, as NMF is NP-hard in general. Under certain restrictions, NMF can be solved exactly [2, 44] but such methods are computationally infeasible. We give efficient algorithms for NMF and show the superior performance empirically.

**Matrix Completion** is another variant of Matrix Factorization arising in recommender systems [35] and bio-informatics [39, 60], which is an active research topic due to the hard non-convex optimization problem [15, 23]. The state-of-the-art methods were proposed in [33, 65] and other recent methods include [66]. Here, our algorithms are either faster or competitive.

Our algorithms are also applicable to Graph Regularized NMF (GNMF) [13], Sparse NMF [8], Nuclear Norm Regularized problems [14, 32], Symmetric NMF via non-symmetric extension [68].

## 2 Matrix Factorization Problem Setting and Algorithms

**Notation.** We refer to [55] for standard notation, unless specified otherwise.

Formally, in a matrix factorization problem, given a matrix $\mathbf{A} \in \mathbb{R}^{M \times N}$, we want to obtain the factors $\mathbf{U} \in \mathbb{R}^{M \times K}$ and $\mathbf{Z}^{K \times N}$ such that $\mathbf{A} \approx \mathbf{UZ}$, which is captured by the following non-convex problem

$$\min_{\mathbf{U} \in \mathcal{U}, \mathbf{Z} \in \mathcal{Z}} \left\{ \Psi(\mathbf{U}, \mathbf{Z}) := \frac{1}{2} \|\mathbf{A} - \mathbf{UZ}\|_F^2 + \mathcal{R}_1(\mathbf{U}) + \mathcal{R}_2(\mathbf{Z}) \right\} , \tag{2.1}$$

where $\mathcal{R}_1(\mathbf{U}) + \mathcal{R}_2(\mathbf{Z})$ is the separable regularization term, $\frac{1}{2} \|\mathbf{A} - \mathbf{UZ}\|_F^2$ is the data-fitting term, and $\mathcal{U}, \mathcal{Z}$ are the constraint sets for $\mathbf{U}$ and $\mathbf{Z}$ respectively. Here, $\mathcal{R}_1(\mathbf{U})$ and $\mathcal{R}_2(\mathbf{Z})$ can be potentially non-convex extended real valued functions and possibly non-smooth. In this paper, we propose to make use of BPG and its inertial variant CoCaIn BPG to solve (2.1). The introduction of these algorithms requires the following preliminary considerations.

**Definition 2.1.** (Kernel Generating Distance [9]) Let $C$ be a nonempty, convex and open subset of $\mathbb{R}^d$. Associated with $C$, a function $h : \mathbb{R}^d \to (-\infty, +\infty]$ is called a *kernel generating distance* if it satisfies: (i) $h$ is proper, lower semicontinuous and convex, with $\operatorname{dom} h \subset \overline{C}$ and $\operatorname{dom} \partial h = C$, and (ii) $h$ is $C^1$ on $\operatorname{int} \operatorname{dom} h \equiv C$. We denote the class of kernel generating distances by $\mathcal{G}(C)$.

For every $h \in \mathcal{G}(C)$, the associated Bregman distance is given by $D_h : \operatorname{dom} h \times \operatorname{int} \operatorname{dom} h \to \mathbb{R}_+$:

$$D_h(x, y) := h(x) - [h(y) + \langle \nabla h(y), x - y \rangle] .$$

For examples, consider the following kernel generating distances:

$$h_0(x) = \frac{1}{2} \|x\|^2 , \quad h_1(x) = \frac{1}{4} \|x\|^4 + \frac{1}{2} \|x\|^2 \quad \text{and} \quad h_2(x) = \frac{1}{3} \|x\|^3 + \frac{1}{2} \|x\|^2 .$$

The Bregman distances associated with $h_0(x)$ is the Euclidean distance. The Bregman distances associated with $h_1$ and $h_2$ appear in the context of non-convex quadratic inverse problems [9, 46] and non-convex cubic regularized problems [46] respectively. For a review on the recent literature, we refer the reader to [59] and for early work on Bregman distances to [16].

These distance measures are key for development of algorithms for the following class of non-convex additive composite problems

$$(\mathcal{P}) \qquad \inf \left\{ \Psi \equiv f(x) + g(x) : x \in \overline{C} \right\}, \tag{2.2}$$

which is assumed to satisfy the following standard assumption [9].

**Assumption A.** (i) $h \in \mathcal{G}(C)$ with $\overline{C} = \overline{\mathrm{dom}\, h}$. (ii) $f : \mathbb{R}^d \to (-\infty, +\infty]$ is a proper and lower semicontinuous function (potentially non-convex) with $\mathrm{dom}\, f \cap C \neq \emptyset$. (iii) $g : \mathbb{R}^d \to (-\infty, +\infty]$ is a proper and lower semicontinuous function (potentially non-convex) with $\mathrm{dom}\, h \subset \mathrm{dom}\, g$, which is continuously differentiable on $C$. (iv) $v(\mathcal{P}) := \inf \left\{ \Psi(x) : x \in \overline{C} \right\} > -\infty$.

**Matrix Factorization Example.** A special case of (2.2) is the following problem,

$$\inf \left\{ \Psi(\mathbf{U}, \mathbf{Z}) := f_1(\mathbf{U}) + f_2(\mathbf{Z}) + g(\mathbf{U}, \mathbf{Z}) : (\mathbf{U}, \mathbf{Z}) \in \overline{C} \right\}. \tag{2.3}$$

We denote $f(\mathbf{U}, \mathbf{Z}) = f_1(\mathbf{U}) + f_2(\mathbf{Z})$. Many practical matrix factorization problems can be cast into the form of (2.1). The choice of $f$ and $g$ is dependent on the problem, for which we provide some examples in Section 3. Here $f_1, f_2$ satisfy the assumptions of $f$ with dimensions chosen accordingly. Moreover by definition, $f$ is separable in $\mathbf{U}$ and $\mathbf{Z}$, which we assume only for practical reasons. Also, the choice of $f, g$ may not be unique. For example, in (2.1) when $\mathcal{R}_1(\mathbf{U}) = \frac{\lambda_0}{2} \|\mathbf{U}\|_F^2$ and $\mathcal{R}_2(\mathbf{Z}) = \frac{\lambda_0}{2} \|\mathbf{Z}\|_F^2$ the choice of $f$ as in (2.3) can be $\mathcal{R}_1 + \mathcal{R}_2$ and $g = \frac{1}{2} \|\mathbf{A} - \mathbf{U}\mathbf{Z}\|_F^2$. However, the other choice is to set $g = \Psi$ and $f := 0$.

## 2.1 BPG-MF: Bregman Proximal Gradient for Matrix Factorization

We require the notion of Bregman Proximal Gradient Mapping [9, Section 3.1] given by

$$T_\lambda(x) = \mathrm{argmin} \left\{ f(u) + \langle \nabla g(x), u - x \rangle + \frac{1}{\lambda} D_h(u, x) : u \in \overline{C} \right\}. \tag{2.4}$$

Then, the update step of Bregman Proximal Gradient (BPG) [9] for solving (2.2) is $x^{k+1} \in T_\lambda(x^k)$, for some $\lambda > 0$ and $h \in \mathcal{G}(C)$. Convergence of BPG relies on a generalized notion of Lipschitz continuity, the so-called $L$-smad property (Defintion 2.2).

**Beyond Lipschitz continuity.** BPG extends upon the popular proximal gradient methods, for which convergence relies on Lipschitz continuity of the smooth part of the objective in (2.2). However, such a notion of Lipschitz continuity is restrictive for many practical applications such as Poisson linear inverse problems [4], quadratic inverse problems [9, 46], cubic regularized problems [46] and robust denoising problems with non-convex total variation regularization [46]. The extensions for generalized notions of Lipschitz continuity of gradients is an active area of research [6, 4, 40, 9]. We consider the following from [9].

**Definition 2.2** ($L$-smad property). The function $g$ is said to be $L$-smooth adaptable ($L$-smad) on $C$ with respect to $h$, if and only if $Lh - g$ and $Lh + g$ are convex on $C$.

When $h(x) = \frac{1}{2} \|x\|^2$, $L$-smad property is implied by Lipschitz continuous gradient. Consider the function $f(x) = x^4$, it is $L$-smad with respect to $h(x) = x^4$ and $L \geq 1$, however $\nabla f$ is not Lipschitz continuous.

Now, we are ready to present the BPG algorithm for Matrix Factorization.

---

**BPG-MF: BPG for Matrix Factorization.**
**Input.** Choose $h \in \mathcal{G}(C)$ with $C \equiv \mathrm{int}\, \mathrm{dom}\, h$ such that $g$ satisfies $L$-smad with respect to $h$ on $C$.
**Initialization.** $(\mathbf{U^1}, \mathbf{Z^1}) \in \mathrm{int}\, \mathrm{dom}\, h$ and let $\lambda > 0$.
**General Step.** For $k = 1, 2, \ldots$, compute

$$\mathbf{P^k} = \lambda \nabla_\mathbf{U} g(\mathbf{U^k}, \mathbf{Z^k}) - \nabla_\mathbf{U} h(\mathbf{U^k}, \mathbf{Z^k}), \quad \mathbf{Q^k} = \lambda \nabla_\mathbf{Z} g(\mathbf{U^k}, \mathbf{Z^k}) - \nabla_\mathbf{Z} h(\mathbf{U^k}, \mathbf{Z^k}),$$

$$(\mathbf{U^{k+1}}, \mathbf{Z^{k+1}}) \in \mathrm{argmin}_{(\mathbf{U}, \mathbf{Z}) \in \overline{C}} \left\{ \lambda f(\mathbf{U}, \mathbf{Z}) + \langle \mathbf{P^k}, \mathbf{U} \rangle + \langle \mathbf{Q^k}, \mathbf{Z} \rangle + h(\mathbf{U}, \mathbf{Z}) \right\}. \tag{2.5}$$

---

Under Assumption A and the following one (mostly satisfied in practice), BPG is well-defined [9].

**Assumption B.** The range of $T_\lambda$ lies in $C$ and, for all $\lambda > 0$, the function $h + \lambda f$ is supercoercive.

The update step for BPG-MF is easy to derive from BPG, however convergence of BPG also relies on the "right" choice of kernel generating distance $h$ and the $L$-smad condition. Finding $h$ such that $L$-smad holds (also see Section 2.2) and that the update step can be given in closed form (also see Section 2.4) is our main contribution and allows us to invoke the convergence results from [9]. The convergence result states that the whole sequence of iterates generated by BPG-MF converges to a stationary point, precisely given in Theorem 2.2. The result depends on the non-smooth KL-property (see [7, 3, 8]) which is a mild requirement and is satisfied by most practical objectives. We provide below the convergence result in [9, Theorem 4.1] adapted to BPG-MF.

**Theorem 2.1** (Global Convergence of BPG-MF). *Let Assumptions A and B hold and let $g$ be $L$-smad with respect to $h$, where $h$ is assumed to be $\sigma$-strongly convex with full domain. Assume $\nabla g, \nabla h$ to be Lipschitz continuous on any bounded subset. Let $\left\{(\mathbf{U^{k+1}}, \mathbf{Z^{k+1}})\right\}_{k \in \mathbb{N}}$ be a bounded sequence generated by BPG-MF with $0 < \lambda L < 1$, and suppose $\Psi$ satisfies the KL property, then, such a sequence has finite length, and converges to a critical point.*

## 2.2 New Bregman Distance for Matrix Factorization

We prove the $L$-smad property for the term $g(\mathbf{U}, \mathbf{Z}) = \frac{1}{2} \|\mathbf{A} - \mathbf{UZ}\|_F^2$ of the matrix factorization problem in (2.1). The kernel generating distance is a linear combination of

$$h_1(\mathbf{U}, \mathbf{Z}) := \left(\frac{\|\mathbf{U}\|_F^2 + \|\mathbf{Z}\|_F^2}{2}\right)^2 \quad \text{and} \quad h_2(\mathbf{U}, \mathbf{Z}) := \frac{\|\mathbf{U}\|_F^2 + \|\mathbf{Z}\|_F^2}{2}, \quad (2.6)$$

and it is designed to also allow for closed form updates (see Section 2.4).

**Proposition 2.1.** *Let $g, h_1, h_2$ be as defined above. Then, for $L \geq 1$, the function $g$ satisfies the $L$-smad property with respect to the following kernel generating distance*

$$h_a(\mathbf{U}, \mathbf{Z}) = 3h_1(\mathbf{U}, \mathbf{Z}) + \|\mathbf{A}\|_F h_2(\mathbf{U}, \mathbf{Z}). \quad (2.7)$$

The proof is given in Section G.1 in the supplementary material. The Bregman distances considered in previous works [46, 9] are separable and not applicable for matrix factorization problems. The inherent coupling between two subsets of variables $\mathbf{U}, \mathbf{Z}$ is the main source of non-convexity in the objective $g$. The kernel generating distance (in particular $h_1$ in (2.7)) contains the interaction/coupling terms between $\mathbf{U}$ and $\mathbf{Z}$ which makes it amenable for matrix factorization problems.

## 2.3 CoCaIn BPG-MF: An Adaptive Inertial Bregman Proximal Gradient Method

The goal of this section is to introduce an inertial variant of BPG-MF, called CoCaIn BPG-MF. The effective step-size choice for BPG-MF can be restrictive due to large constant like $\|\mathbf{A}\|_F$ (see (2.7)), for which we present a practical example in the numerical experiments. In order to allow for larger step-sizes, one needs to adapt it locally, which is often done via a backtracking procedure. CoCaIn BPG-MF combines inertial steps with a novel backtracking procedure proposed in [46].

Inertial algorithms often lead to better convergence [51, 53, 46]. The classical Nesterov Accelerated Gradient (NAG) method [47] and the popular Fast Iterative Shrinkage-Thresholding Algorithm (FISTA) [5] employ an extrapolation based inertial strategy. However, the extrapolation is governed by a parameter which is typically scheduled to follow certain iteration-dependent scheme [47, 29]and is restricted to the convex setting. Recently with Convex–Concave Inertial Bregman Proximal Gradient (CoCaIn BPG) [46], it was shown that one could leverage the upper bound (convexity of $Lh - g$) and lower bound (convexity of $Lh + g$) to incorporate inertia in an adaptive manner.

We recall now the update steps of CoCaIn BPG [46] to solve (2.2). Let $h \in \mathcal{G}(C)$, $\lambda > 0$, and $x^0 = x^1 \in \mathbb{R}^d$ be an initalization, then in each iteration the extrapolated point $y^k = x^k + \gamma_k(x^k - x^{k-1})$ is computed followed by a BPG like update (at $y^k$) given by $x^{k+1} \in T_{\tau_k}(y^k)$, where $\gamma_k$ is the inertial parameter and $\tau_k$ is the step-size parameter. Similar conditions to BPG are required for the convergence to a stationary point. We use CoCaIn BPG for Matrix Factorization (CoCaIn BPG-MF) and our proposed novel kernel generating distance $h$ from (2.7) makes the convergence results of [46] applicable. Along with Assumption B, we require the following assumption.

**Assumption C.** (i) There exists $\alpha \in \mathbb{R}$ such that $f(\mathbf{U}, \mathbf{Z}) - \frac{\alpha}{2}\left(\|\mathbf{U}\|_F^2 + \|\mathbf{Z}\|_F^2\right)$ is convex.

(ii) The kernel generating distance $h$ is $\sigma$-strongly convex on $\mathbb{R}^{M \times K} \times \mathbb{R}^{K \times N}$.

The Assumption C(i) refers to notion of semi-convexity of the function $f$, (see [50, 46]) and seems to be closely connected to the inertial feature of an algorithm. For notational brevity, we use $D_g(x, y) := g(x) - [g(y) + \langle \nabla g(y), x - y\rangle]$ which may also be negative if g is not a kernel generating distance. Moreover, we use $D_h((\mathbf{X}_1, \mathbf{Y}_1), (\mathbf{X}_2, \mathbf{Y}_2))$ as $D_h(\mathbf{X}_1, \mathbf{Y}_1, \mathbf{X}_2, \mathbf{Y}_2)$. We provide CoCaIn BPG-MF below.

---

**CoCaIn BPG-MF: Convex–Concave Inertial BPG for Matrix Factorization.**

**Input.** Choose $\delta, \varepsilon > 0$ with $1 > \delta > \epsilon$, $h \in \mathcal{G}(C)$ with $C \equiv \text{int dom } h$, g is $L$-smad on $C$ w.r.t $h$.

**Initialization.** $(\mathbf{U^1}, \mathbf{Z^1}) = (\mathbf{U^0}, \mathbf{Z^0}) \in \text{int dom } h \cap \text{dom } f$, $\bar{L}_0 > \frac{-\alpha}{(1-\delta)\sigma}$ and $\tau_0 \leq \bar{L}_0^{-1}$.

**General Step.** For $k = 1, 2, \ldots$, compute extrapolated points

$$Y_{\mathbf{U}}^{\mathbf{k}} = \mathbf{U}^k + \gamma_k\left(\mathbf{U^k} - \mathbf{U^{k-1}}\right) \quad \text{and} \quad Y_{\mathbf{Z}}^{\mathbf{k}} = \mathbf{Z}^k + \gamma_k\left(\mathbf{Z^k} - \mathbf{Z^{k-1}}\right), \qquad (2.8)$$

where $\gamma_k \geq 0$ such that

$$(\delta - \varepsilon)D_h\left(\mathbf{U^{k-1}}, \mathbf{Z^{k-1}}, \mathbf{U^k}, \mathbf{Z^k}\right) \geq (1 + \underline{L}_k \tau_{k-1})D_h\left(\mathbf{U^k}, \mathbf{Z^k}, Y_{\mathbf{U}}^{\mathbf{k}}, Y_{\mathbf{Z}}^{\mathbf{k}}\right), \qquad (2.9)$$

where $\underline{L}_k$ satisfies

$$D_g\left(\mathbf{U^k}, \mathbf{Z^k}, Y_{\mathbf{U}}^{\mathbf{k}}, Y_{\mathbf{Z}}^{\mathbf{k}}\right) \geq -\underline{L}_k D_h\left(\mathbf{U^k}, \mathbf{Z^k}, Y_{\mathbf{U}}^{\mathbf{k}}, Y_{\mathbf{Z}}^{\mathbf{k}}\right). \qquad (2.10)$$

Choose $\bar{L}_k \geq \bar{L}_{k-1}$, and set $\tau_k \leq \min\{\tau_{k-1}, \bar{L}_k^{-1}\}$. Now, compute

$$\mathbf{P^k} = \tau_k \nabla_{\mathbf{U}} g\left(Y_{\mathbf{U}}^{\mathbf{k}}, Y_{\mathbf{Z}}^{\mathbf{k}}\right) - \nabla_{\mathbf{U}} h(Y_{\mathbf{U}}^{\mathbf{k}}, Y_{\mathbf{Z}}^{\mathbf{k}}), \quad \mathbf{Q^k} = \tau_k \nabla_{\mathbf{Z}} g\left(Y_{\mathbf{U}}^{\mathbf{k}}, Y_{\mathbf{Z}}^{\mathbf{k}}\right) - \nabla_{\mathbf{Z}} h(Y_{\mathbf{U}}^{\mathbf{k}}, Y_{\mathbf{Z}}^{\mathbf{k}}),$$

$$(\mathbf{U^{k+1}}, \mathbf{Z^{k+1}}) \in \underset{(\mathbf{U}, \mathbf{Z}) \in \overline{C}}{\operatorname{argmin}} \left\{\tau_k f(\mathbf{U}, \mathbf{Z}) + \langle \mathbf{P^k}, \mathbf{U}\rangle + \langle \mathbf{Q^k}, \mathbf{Z}\rangle + h(\mathbf{U}, \mathbf{Z})\right\}, \qquad (2.11)$$

such that $\bar{L}_k$ satisfies

$$D_g\left(\mathbf{U^{k+1}}, \mathbf{Z^{k+1}}, Y_{\mathbf{U}}^{\mathbf{k}}, Y_{\mathbf{Z}}^{\mathbf{k}}\right) \leq \bar{L}_k D_h\left(\mathbf{U^{k+1}}, \mathbf{Z^{k+1}}, Y_{\mathbf{U}}^{\mathbf{k}}, Y_{\mathbf{Z}}^{\mathbf{k}}\right). \qquad (2.12)$$

---

The extrapolation step is performed in (2.8), which is similar to NAG/FISTA. However, the inertia cannot be arbitrary and the analysis from [46] requires step (2.9) which is governed by the convexity of lower bound, $\underline{L}_k h + g$, however only locally as in (2.10). The update step (2.11) is similar to BPG-MF, however the step-size is controlled via the convexity of upper bound $\bar{L}_k h - g$, but only locally as in (2.12). The local adaptation of the steps (2.10) and (2.12) is performed via backtracking. Since, $\bar{L}_k$ can be potentially very small compared to $L$, hence potentially large steps can be taken. There is no restriction on $\underline{L}_k$ in each iteration, and smaller $\underline{L}_k$ can result in high value for the inertial parameter $\gamma_k$. Thus the algorithm in essence aims to detect "local convexity" of the objective. The update steps of CoCaIn BPG-MF can be executed sequentially without any nested loops for the backtracking. One can always find the inertial parameter $\gamma_k$ in (2.9) due to [46, Lemma 4.1]. For certain cases, (2.9) yields an explicit condition on $\gamma_k$. For example, for $h(\mathbf{U}, \mathbf{Z}) = \frac{1}{2}(\|\mathbf{U}\|_F^2 + \|\mathbf{Z}\|_F^2)$, we have $0 \leq \gamma_k \leq \sqrt{\frac{\delta - \varepsilon}{1 + \tau_{k-1}\underline{L}_k}}$. We now provide below the convergence result from [46, Theorem 5.2] adapted to CoCaIn BPG-MF.

**Theorem 2.2** (Global Convergence of CoCaIn BPG-MF). *Let Assumptions A, B and C hold, let g be $L$-smad with respect to $h$ with full domain. Assume $\nabla g, \nabla h$ to be Lipschitz continuous on any bounded subset. Let $\left\{(\mathbf{U^{k+1}}, \mathbf{Z^{k+1}})\right\}_{k \in \mathbb{N}}$ be a bounded sequence generated by CoCaIn BPG-MF, and suppose $f, g$ satisfy the KL property, then, such a sequence has finite length, and converges to a critical point.*

## 2.4 Closed Form Solutions for Update Steps of BPG-MF and CoCaIn BPG-MF

Our second significant contribution is to make BPG-MF and CoCaIn BPG-MF an efficient choice for solving Matrix Factorization, namely closed form expressions for the main update steps (2.5), (2.11). For the derivation, we refer to the supplementary material, here we just state our results.

For the L2-regularized problem

$$g(\mathbf{U}, \mathbf{Z}) = \frac{1}{2} \|\mathbf{A} - \mathbf{U}\mathbf{Z}\|_F^2, \quad f(\mathbf{U}, \mathbf{Z}) = \frac{\lambda_0}{2} \left( \|\mathbf{U}\|_F^2 + \|\mathbf{Z}\|_F^2 \right), \quad h = h_a$$

with $c_1 = 3, c_2 = \|\mathbf{A}\|_F$ and $0 < \lambda < 1$ the BPG-MF updates are:

$$\mathbf{U^{k+1}} = -r\mathbf{P^k}, \mathbf{Z^{k+1}} = -r\mathbf{Q^k} \text{ with } r \geq 0, c_1 \big( \left\| -\mathbf{P^k} \right\|_F^2 + \left\| -\mathbf{Q^k} \right\|_F^2 \big) r^3 + (c_2 + \lambda_0) r - 1 = 0.$$

For NMF with additional non-negativity constraints, we replace $-\mathbf{P^k}$ and $-\mathbf{Q^k}$ by $\Pi_+(-\mathbf{P^k})$ and $\Pi_+(-\mathbf{Q^k})$ respectively where $\Pi_+(.) = \max\{0, .\}$ and $\max$ is applied element wise.

Now consider the following L1-Regularized problem

$$g(\mathbf{U}, \mathbf{Z}) = \frac{1}{2} \|\mathbf{A} - \mathbf{U}\mathbf{Z}\|_F^2, \quad f(\mathbf{U}, \mathbf{Z}) = \lambda_1 \left( \|\mathbf{U}\|_1 + \|\mathbf{Z}\|_1 \right), \quad h = h_a. \tag{2.13}$$

The soft-thresholding operator is defined for any $y \in \mathbb{R}^d$ by $\mathcal{S}_\theta(y) = \max\{|y| - \theta, 0\} \operatorname{sgn}(y)$ where $\theta > 0$. Set $c_1 = 3, c_2 = \|\mathbf{A}\|_F$ and $0 < \lambda < 1$ the BPG-MF updates with the above given $g, f, h$ are:

$$\mathbf{U^{k+1}} = r\mathcal{S}_{\lambda_1\lambda}(-\mathbf{P^k}), \mathbf{Z^{k+1}} = r\mathcal{S}_{\lambda_1\lambda}(-\mathbf{Q^k}) \text{ with } r \geq 0 \text{ and}$$

$$c_1 \left( \left\| \mathcal{S}_{\lambda_1\lambda}(-\mathbf{P^k}) \right\|_F^2 + \left\| \mathcal{S}_{\lambda_1\lambda}(-\mathbf{Q^k}) \right\|_F^2 \right) r^3 + c_2 r - 1 = 0.$$

We denote a vector of ones as $\mathbf{e}_D \in \mathbb{R}^D$. For additional non-negativity constraints we need to replace $\mathcal{S}_{\lambda_1\lambda}(-\mathbf{P^k})$ with $\Pi_+(-(\mathbf{P^k} + \lambda_1\lambda\mathbf{e}_M\mathbf{e}_K^T))$ and $\mathcal{S}_{\lambda_1\lambda}(-\mathbf{Q^k})$ to $\Pi_+(-(\mathbf{Q^k} + \lambda_1\lambda\mathbf{e}_K\mathbf{e}_N^T))$. Excluding the gradient computation, the computational complexity of our updates is $O(MK + NK)$ only, thanks to linear operations. PALM and iPALM additionally involve calculating Lipschitz constants with at most $O(K^2 \max\{M, N\}^2)$ computations. Examples like Graph Regularized NMF (GNMF) [13], Sparse NMF [8], Matrix Completion [35], Nuclear Norm Regularization [14, 32], Symmetric NMF [68] and proofs are given in the supplementary material.

## 3 Experiments

In this section, we show experiments for (2.1). Denote the regularization settings, **R1:** with $\mathcal{R}_1 \equiv \mathcal{R}_2 \equiv 0$, **R2:** with L2 regularization $\mathcal{R}_1(\mathbf{U}) = \frac{\lambda_0}{2} \|\mathbf{U}\|_F^2$ and $\mathcal{R}_2(\mathbf{Z}) = \frac{\lambda_0}{2} \|\mathbf{Z}\|_F^2$ for some $\lambda_0 > 0$, **R3:** with L1 Regularization $\mathcal{R}_1(\mathbf{U}) = \lambda_0 \|\mathbf{U}\|_1$ and $\mathcal{R}_2(\mathbf{Z}) = \lambda_0 \|\mathbf{Z}\|_1$ for some $\lambda_0 > 0$.

**Algorithms.** We compare our first order optimization algorithms, BPG-MF and CoCaIn BPG-MF, and recent state-of-the-art optimization methods iPALM [53] and PALM [8]. We focus on algorithms that guarantee convergence to a stationary point. We also use BPG-MF-WB, where WB stands for "with backtracking", which is equivalent to CoCaIn BPG-MF with $\gamma_k \equiv 0$. We use two settings for iPALM, where all the extrapolation parameters are set to a single value $\beta$ set to $0.2$ and $0.4$. PALM is equivalent to iPALM if $\beta = 0$. We use the same initialization for all methods.

**Simple Matrix Factorization.** We set $\mathcal{U} = \mathbb{R}^{M \times K}$ and $\mathcal{Z} = \mathbb{R}^{K \times N}$. We use a randomly generated synthetic data matrix with $A \in \mathbb{R}^{200 \times 200}$ and report performance in terms of function value for three regularization settings, **R1**, **R2** and **R3** with $K = 5$. Note that this enforces a factorization into at most rank 5 matrizes $\mathbf{U}$ and $\mathbf{Z}$, which yields an additional implicit regularization. For **R2** and **R3** we use $\lambda_0 = 0.1$. CoCaIn BPG-MF is superior[1] as shown in Figure 1 .

**Statistical Evaluation.** We also provide the statistical evaluation of all the algorithms in Figure 2, for the above problem. The optimization variables are sampled from [0,0.1] and 50 random seeds are considered. CoCaIn BPG outperforms other methods, however PALM methods are also very competitive. In L1 regularization setting, the performance of CoCaIn BPG is the best. In all settings, BPG-MF performance is worst due to a constant step size, which might change in settings where local adapation with backtracking line search is computationally not feasible.

**Matrix Completion.** In recommender systems [35] given a matrix $A$ with entries at few index pairs in set $\Omega$, the goal is to obtain factors $\mathbf{U}$ and $\mathbf{Z}$ that generalize via following optimization problem

$$\min_{\mathbf{U} \in \mathbb{R}^{M \times K}, \mathbf{Z} \in \mathbb{R}^{K \times N}} \left\{ \Psi(\mathbf{U}, \mathbf{Z}) := \frac{1}{2} \|P_\Omega(\mathbf{A} - \mathbf{U}\mathbf{Z})\|_F^2 + \frac{\lambda_0}{2} \left( \|\mathbf{U}\|_F^2 + \|\mathbf{Z}\|_F^2 \right) \right\}, \tag{3.1}$$

where $P_\Omega$ preserves the given matrix entries and sets others to zero. We use 80% data of MovieLens-100K, MovieLens-1M and MovieLens-10M [30] datasets and use other 20% to test (details in the supplementary material). CoCaIn BPG-MF is faster than all methods as given in Figure 3.

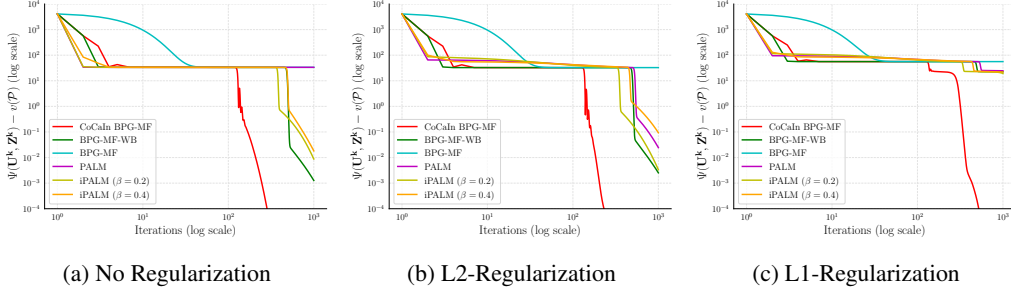

(a) No Regularization    (b) L2-Regularization    (c) L1-Regularization

Figure 1: **Simple Matrix Factorization on Synthetic Dataset.**

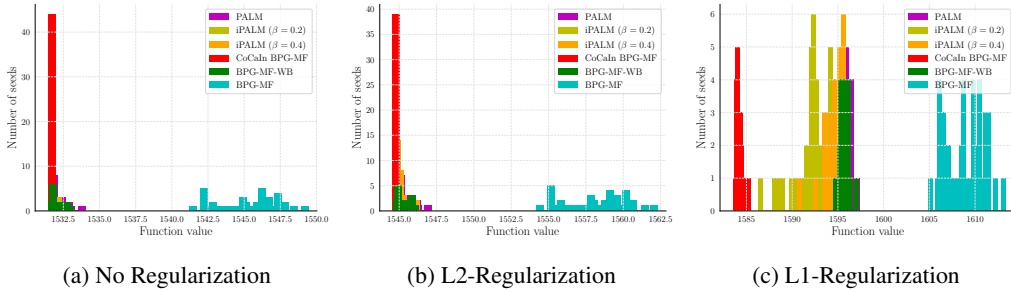

(a) No Regularization    (b) L2-Regularization    (c) L1-Regularization

Figure 2: **Statistical Evaluation on Simple Matrix Factorization.**

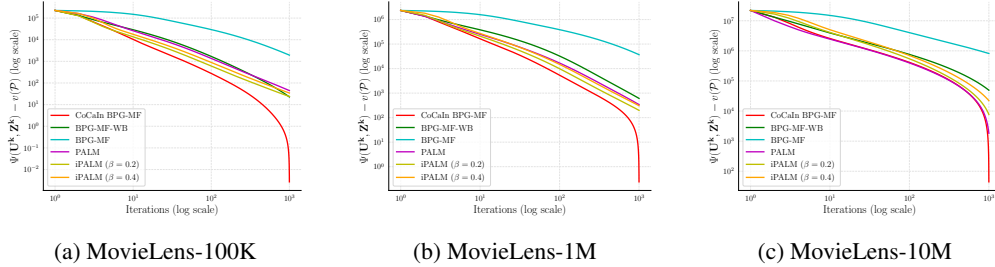

(a) MovieLens-100K    (b) MovieLens-1M    (c) MovieLens-10M

Figure 3: **Matrix Completion on MovieLens Datasets [30].**

As evident from Figures 1, 4, 3, CoCaIn BPG-MF, BPG-MF-WB can result in better performance than well known alternating methods. BPG-MF is not better than PALM and iPALM because of prohibitively small step-sizes (due to $\|\mathbf{A}\|_F$ in (2.7)), which is resolved by CoCaIn BPG-MF and BPG-MF-WB using backtracking. Time comparisons are provided in the supplementary material, where we show that our methods are competitive.

## Conclusion and Extensions

We proposed non-alternating algorithms to solve matrix factorization problems, contrary to the typical alternating strategies. We use the Bregman proximal algorithms, BPG [9] and an inertial variant CoCaIn BPG [46] for matrix factorization problems. We developed a novel Bregman distance, crucial for proving convergence to a stationary point. Moreover, we also provide non-trivial efficient closed form update steps for many matrix factorization problems. This line of thinking raises new open questions, such as extensions to Tensor Factorization [34], to Robust Matrix Factorization [65], stochastic variants [20, 27, 45, 48] and state-of-the-art matrix factorization model [33].

## Footnotes

[1]Note that in the $y$-axis label $v(\mathcal{P})$ is the least objective value attained by any of the methods.

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
