[Supplementary Material]

# A Discussion

We briefly remark some properties of the update steps of BPG-methods. Note that the updates are independent for $\mathbf{U}$ and $\mathbf{Z}$ in (1.3), where updates can be done in parallel blockwise (communication is only required to solve the 1D cubic equation). This can be potentially used to increase the speedup in practice, in particular for large matrices. Some terms in gradients overlap, so using temporary variables in implementation can possibly increase the speedup. These speedups are not restricted to (1.3), however to all the update steps we mentioned in this paper.

We now provide insights on why BPG-methods are a better choice over other methods, with focus on alternating methods.

- PALM-methods estimate a Lipschitz constant with respect to a block of coordinates in each iteration, which is expensive for large block matrices. BPG-methods use a global L-smad constant, which is computed only once.

- PALM-methods cannot be parallelized block wise, for example, in the two block case, the computation of the Lipschitz constant of the second block must wait for the first block to be updated, hence it is inherently serial.

- Alternating minimization methods do not converge for non-smooth regularization terms and can be inefficient (for, e.g., ALS) for some matrix factorization problems (see, for example, [34, 54]). BPG-methods and PALM-methods converge (due to linearization).

- PALM is not applicable to the 2D function $g(x, y) = (x^3 + y^3)^2$, because the block-wise Lipschitz continuity of the gradients fails to hold even after fixing one variable. BPG-methods are applicable here.

- PALM is not applicable to, for example, symmetric matrix Factorization as also pointed in [21] or the following penalty method based (relaxed) orthogonal NMF problem (see (1.1))

$$\min_{\mathbf{U} \in \mathcal{U}, \mathbf{Z} \in \mathcal{Z}} \left\{ \Psi \equiv \frac{1}{2} \left\| \mathbf{A} - \mathbf{UZ} \right\|_F^2 + \frac{\rho}{2} \left\| \mathbf{U}^T \mathbf{U} - \mathbf{I} \right\|_F^2 + \mathbf{I}_{\mathbf{U} \geq \mathbf{0}} + \mathbf{I}_{\mathbf{Z} \geq \mathbf{0}} + \mathcal{R}_1(\mathbf{U}) + \mathcal{R}_2(\mathbf{Z}) \right\} ,$$

  where second term does not have a block-wise Lipschitz continuous gradient for any $\rho > 0$. Here BPG-methods are applicable (similarly also for Projective NMF) with minor changes to the Bregman distance. For symmetric matrix factorization, we recover the kernel generating distances proposed in [21].

- BPG-methods are very general so the choice of applications will increase substantially and this will potentially open doors to design new losses and regularizers, without restricting to Lipschitz continuous gradients.

**State of the art models.** The state-of-the-art matrix factorization models in [33] go beyond two factors and new factorization models are introduced. BPG algorithms are not valid in their setting, and requires potentially developing new Bregman distances. Also, BPG based methods are not applicable for big data setting, where stochasticity plays a major role. The stochastic version of BPG was recently proposed in [20]. The empirical comparisons to [33] is still open. Moreover, designing the appropriate kernels in the context of new factorization models can possibly require substantially technical proofs.

**Extensions.** Our algorithms can potentially extended to several applications, for example, multi-task learning, general matrix sensing, weighted PCA with various applications including cluster analysis, phase retrieval, power system state estimation. Even though CoCaIn BPG-MF appears to perform best, the performance of BPG-MF which forms the basis for CoCaIn BPG-MF, is worst as illustrated in 3. This possibly implies that the kernel choice or the coefficients involved in the kernels are not optimal. Such optimal choice of kernel generating distances were partially explored in the context of symmetric matrix factorization setting in [21], where new Bregman distances based on Gram kernels were introduced with state of the art performance in applicable settings.

# B Overview of the Results

Below, we provide a table with the problem or content description and corresponding section where the results are presented.

## C    Closed Form Solutions Part I for Matrix Factorization

Since, the update steps of BPG-MF and CoCaIn BPG-MF have same structure, we provide the closed form expressions to just BPG-MF. We start with the following technical lemma.

**Lemma C.1.** *Let* $\mathbf{Q} \in \mathbb{R}^{A \times B}$ *for some positive integers $A$ and $B$. Let $t \geq 0$ and $\|\mathbf{Q}\|_F \neq 0$ then*

$$\min_{\mathbf{X} \in \mathbb{R}^{A \times B}} \left\{ \langle \mathbf{Q}, \mathbf{X} \rangle : \|\mathbf{X}\|_F^2 = t^2 \right\} \equiv \min_{\mathbf{X} \in \mathbb{R}^{A \times B}} \left\{ \langle \mathbf{Q}, \mathbf{X} \rangle : \|\mathbf{X}\|_F^2 \leq t^2 \right\} = -t \|\mathbf{Q}\|_F \ ,$$

*with the minimizer at* $\mathbf{X}^* = -t\mathbf{Q}/\|\mathbf{Q}\|_F$.

*Proof.* The proof is inspired from [41, Lemma 9]. On rewriting we have the following equivalence

$$\min_{\mathbf{X} \in \mathbb{R}^{A \times B}} \left\{ \langle \mathbf{Q}, \mathbf{X} \rangle : \|\mathbf{X}\|_F^2 \leq t^2 \right\} \equiv - \max_{\mathbf{X} \in \mathbb{R}^{A \times B}} \left\{ \langle -\mathbf{Q}, \mathbf{X} \rangle : \|\mathbf{X}\|_F^2 \leq t^2 \right\} \ .$$

The expression $\langle -\mathbf{Q}, \mathbf{X} \rangle$ is maximized at $\mathbf{X}^* = c(-\mathbf{Q})$ for certain constant $c$. On substituting we have

$$\langle -\mathbf{Q}, \mathbf{X}^* \rangle = c \|\mathbf{Q}\|_F^2 \ .$$

Since, the dependence on $c$ is linear and we additionally require $\|\mathbf{X}\|_F^2 \leq t^2$, we can set $c = \frac{t}{\|\mathbf{Q}\|_F}$ if $\|\mathbf{Q}\|_F \neq 0$ else $c = 0$. Hence, the minimizer to

$$\min_{\mathbf{X} \in \mathbb{R}^{A \times B}} \left\{ \langle \mathbf{Q}, \mathbf{X} \rangle : \|\mathbf{X}\|_F^2 \leq t^2 \right\}$$

is attained at $\mathbf{X}^* = -t\frac{\mathbf{Q}}{\|\mathbf{Q}\|_F}$ for $\|\mathbf{Q}\|_F \neq 0$ else $\mathbf{X}^* = 0$. The equivalence in the statement follows as $\|\mathbf{X}^*\|_F^2 = t^2$. □

Consider the following non-convex matrix factorization problem

$$\min_{\mathbf{U} \in \mathbb{R}^{M \times K}, \mathbf{Z} \in \mathbb{R}^{K \times N}} \left\{ \Psi(\mathbf{U}, \mathbf{Z}) := \frac{1}{2} \|\mathbf{A} - \mathbf{U}\mathbf{Z}\|_F^2 \right\} \ . \tag{C.1}$$

Denote $g = \Psi$, $f := 0$, $h = h_a$.

**Proposition C.1.** *In BPG-MF, with above defined $g, f, h$ the update steps in each iteration are given by* $\mathbf{U}^{k+1} = -r\,\mathbf{P}^k$, $\mathbf{Z}^{k+1} = -r\,\mathbf{Q}^k$ *where $r$ is the non-negative real root of*

$$c_1 \left( \|\mathbf{Q}^k\|_F^2 + \|\mathbf{P}^k\|_F^2 \right) r^3 + c_2 r - 1 = 0 \,, \tag{C.2}$$

*with $c_1 = 3$ and $c_2 = \|\mathbf{A}\|_F$.*

*Proof.* Consider the following subproblem

$$(\mathbf{U^{k+1}}, \mathbf{Z^{k+1}}) \in \underset{(\mathbf{U},\mathbf{Z}) \in \mathbb{R}^{M \times K} \times \mathbb{R}^{K \times N}}{\operatorname{argmin}} \left\{ \langle \mathbf{P^k}, \mathbf{U} \rangle + \langle \mathbf{Q^k}, \mathbf{Z} \rangle \right.$$

$$\left. + c_1 \left( \frac{\|\mathbf{U}\|_F^2 + \|\mathbf{Z}\|_F^2}{2} \right)^2 + c_2 \left( \frac{\|\mathbf{U}\|_F^2 + \|\mathbf{Z}\|_F^2}{2} \right) \right\}.$$

Denote the objective in the above minimization problem as $\mathcal{O}(\mathbf{U^k}, \mathbf{Z^k})$. Now, the following holds

$$\min_{(\mathbf{U},\mathbf{Z}) \in \mathbb{R}^{M \times K} \times \mathbb{R}^{K \times N}} \left( \mathcal{O}(\mathbf{U^k}, \mathbf{Z^k}) \right)$$

$$\equiv \min_{t_1 \geq 0, t_2 \geq 0} \left\{ \min_{(\mathbf{U},\mathbf{Z}) \in \mathbb{R}^{M \times K} \times \mathbb{R}^{K \times N}, \|\mathbf{U}\|_F = t_1, \|\mathbf{Z}\|_F = t_2} \left( \mathcal{O}(\mathbf{U^k}, \mathbf{Z^k}) \right) \right\}, \quad \text{(C.3)}$$

$$\equiv \min_{t_1 \geq 0, t_2 \geq 0} \left\{ \min_{(\mathbf{U},\mathbf{Z}) \in \mathbb{R}^{M \times K} \times \mathbb{R}^{K \times N}, \|\mathbf{U}\|_F \leq t_1, \|\mathbf{Z}\|_F \leq t_2} \left( \mathcal{O}(\mathbf{U^k}, \mathbf{Z^k}) \right) \right\}, \quad \text{(C.4)}$$

where the first step is a simple rewriting of the objective. The second step is non-trivial. In order to prove (C.4) we rewrite (C.3) as

$$\min_{t_1 \geq 0, t_2 \geq 0} \left\{ \min_{\mathbf{U_1} \in \mathbb{R}^{M \times K}} \left\{ \langle \mathbf{P^k}, \mathbf{U_1} \rangle : \|\mathbf{U_1}\|_F^2 = t_1 \right\} \right.$$

$$+ \min_{\mathbf{Z_1} \in \mathbb{R}^{K \times N}} \left\{ \langle \mathbf{Q^k}, \mathbf{Z_1} \rangle : \|\mathbf{Z_1}\|_F^2 = t_2 \right\}$$

$$\left. + c_1 \left( \frac{t_1^2 + t_2^2}{2} \right)^2 + c_2 \left( \frac{t_1^2 + t_2^2}{2} \right) \right\}.$$

Now, note the following equivalence due to Lemma C.1

$$\min_{\mathbf{U_1} \in \mathbb{R}^{M \times K}} \left\{ \langle \mathbf{P^k}, \mathbf{U_1} \rangle : \|\mathbf{U_1}\|_F^2 = t_1 \right\} \equiv \min_{\mathbf{U_1} \in \mathbb{R}^{M \times K}} \left\{ \langle \mathbf{P^k}, \mathbf{U_1} \rangle : \|\mathbf{U_1}\|_F^2 \leq t_1 \right\},$$

$$\min_{\mathbf{Z_1} \in \mathbb{R}^{K \times N}} \left\{ \langle \mathbf{Q^k}, \mathbf{Z_1} \rangle : \|\mathbf{Z_1}\|_F^2 = t_2 \right\} \equiv \min_{\mathbf{Z_1} \in \mathbb{R}^{K \times N}} \left\{ \langle \mathbf{Q^k}, \mathbf{Z_1} \rangle : \|\mathbf{Z_1}\|_F^2 \leq t_2 \right\}.$$

This proves (C.4). Now, we solve for $(\mathbf{U^{k+1}}, \mathbf{Z^{k+1}})$ via the following strategy. Denote

$$\mathbf{U_1^*}(t_1) \in \operatorname{argmin} \left\{ \langle \mathbf{P^k}, \mathbf{U_1} \rangle : \mathbf{U_1} \in \mathbb{R}^{M \times K}, \|\mathbf{U_1}\|_F^2 \leq t_1 \right\},$$

$$\mathbf{Z_1^*}(t_2) \in \operatorname{argmin} \left\{ \langle \mathbf{Q^k}, \mathbf{Z_1} \rangle : \mathbf{Z_1} \in \mathbb{R}^{K \times N}, \|\mathbf{Z_1}\|_F^2 \leq t_2 \right\}.$$

Then we obtain $(\mathbf{U^{k+1}}, \mathbf{Z^{k+1}}) = (\mathbf{U_1^*}(t_1^*), \mathbf{Z_1^*}(t_2^*))$, where $t_1^*$ and $t_2^*$ are obtained by solving the following two dimensional subproblem

$$(t_1^*, t_2^*) \in \underset{t_1 \geq 0, t_2 \geq 0}{\operatorname{argmin}} \left\{ \min_{\mathbf{U_1} \in \mathbb{R}^{M \times K}} \left\{ \langle \mathbf{P^k}, \mathbf{U_1} \rangle : \|\mathbf{U_1}\|_F^2 \leq t_1 \right\} \right.$$

$$+ \min_{\mathbf{Z_1} \in \mathbb{R}^{K \times N}} \left\{ \langle \mathbf{Q^k}, \mathbf{Z_1} \rangle : \|\mathbf{Z_1}\|_F^2 \leq t_2 \right\}$$

$$\left. + c_1 \left( \frac{t_1^2 + t_2^2}{2} \right)^2 + c_2 \left( \frac{t_1^2 + t_2^2}{2} \right) \right\}.$$

Note that inner minimization subproblems can be trivially solved once we obtain $\mathbf{U_1^*}(t_1)$ and $\mathbf{Z_1^*}(t_2)$ via Lemma C.1. Then the solution to the subproblem in each iteration is as follows:

$$\mathbf{U^{k+1}} = \begin{cases} t_1^* \frac{-\mathbf{P^k}}{\|\mathbf{P^k}\|_F}, & \text{for } \|\mathbf{P^k}\|_F \neq 0, \\ \mathbf{0} & otherwise. \end{cases}$$

$$\mathbf{Z^{k+1}} = \begin{cases} t_2^* \frac{-\mathbf{Q^k}}{\|\mathbf{Q^k}\|_F}, & \text{for } \|\mathbf{Q^k}\|_F \neq 0, \\ \mathbf{0} & otherwise. \end{cases}$$

We solve for $t_1^*$ and $t_2^*$ with the following two dimensional minimization problem

$$\underset{t_1 \geq 0, t_2 \geq 0}{\operatorname{argmin}} \left\{ -t_1 \left\| \mathbf{P^k} \right\|_F - t_2 \left\| \mathbf{Q^k} \right\|_F + c_1 \left( \frac{t_1^2 + t_2^2}{2} \right)^2 + c_2 \left( \frac{t_1^2 + t_2^2}{2} \right) \right\} .$$

Thus, the solutions $t_1^*$ and $t_2^*$ are the non-negative real roots of the following equations

$$- \left\| \mathbf{P^k} \right\|_F + c_1(t_1^2 + t_2^2)t_1 + c_2 t_1 = 0$$
$$- \left\| \mathbf{Q^k} \right\|_F + c_1(t_1^2 + t_2^2)t_2 + c_2 t_2 = 0$$

Further simplifications lead to $t_1 = r \left\| \mathbf{P^k} \right\|_F$ and $t_2 = r \left\| \mathbf{Q^k} \right\|_F$ for some $r \geq 0$ such that $r$ satisfies the following cubic equation

$$c_1 \left( \left\| \mathbf{Q^k} \right\|_F^2 + \left\| \mathbf{P^k} \right\|_F^2 \right) r^3 + c_2 r - 1 = 0 .$$

$\square$

## C.1 Extensions to L2-Regularized Matrix Factorization

We consider the following L2-Regularized Matrix Factorization problem [38].

$$\min_{\mathbf{U} \in \mathbb{R}^{M \times K}, \mathbf{Z} \in \mathbb{R}^{K \times N}} \left\{ \Psi(\mathbf{U}, \mathbf{Z}) := \frac{1}{2} \left\| \mathbf{A} - \mathbf{UZ} \right\|_F^2 + \frac{\lambda_0}{2} \left( \left\| \mathbf{U} \right\|_F^2 + \left\| \mathbf{Z} \right\|_F^2 \right) \right\} . \quad \text{(C.5)}$$

Denote $g := \frac{1}{2} \left\| \mathbf{A} - \mathbf{UZ} \right\|_F^2$, $f := \frac{\lambda_0}{2} \left( \left\| \mathbf{U} \right\|_F^2 + \left\| \mathbf{Z} \right\|_F^2 \right)$ and $h = h_a$.

**Proposition C.2.** *In BPG-MF, with the above defined $g, f, h$ the update steps in each iteration are given by $\mathbf{U^{k+1}} = -r \mathbf{P^k}$, $\mathbf{Z^{k+1}} = -r \mathbf{Q^k}$ where $r$ is the non-negative real root of*

$$c_1 \left( \left\| \mathbf{Q^k} \right\|_F^2 + \left\| \mathbf{P^k} \right\|_F^2 \right) r^3 + (c_2 + \lambda_0)r - 1 = 0, \quad \text{(C.6)}$$

*with $c_1 = 3$ and $c_2 = \left\| \mathbf{A} \right\|_F$.*

We skip the proof as it is very similar to Proposition C.1 and only change is in $c_2$.

## C.2 Extensions to Graph Regularized Matrix Factorization

Graph Regularized Matrix Factorization was proposed in [13]. However, they used non-negativity constraints. We simplify the problem here by not considering the non-negativity constraints. We later show in Section D.3, how the non-negativity constraints are handled. Here, given $\mathcal{L} \in \mathbb{R}^{M \times M}$ we are interested to solve

$$\min_{\mathbf{U} \in \mathbb{R}^{M \times K}, \mathbf{Z} \in \mathbb{R}^{K \times N}} \left\{ \Psi(\mathbf{U}, \mathbf{Z}) := \frac{1}{2} \left\| \mathbf{A} - \mathbf{UZ} \right\|_F^2 + \frac{\mu_0}{2} \operatorname{tr}(\mathbf{U}^T \mathcal{L} \mathbf{U}) + \frac{\lambda_0}{2} \left( \left\| \mathbf{U} \right\|_F^2 + \left\| \mathbf{Z} \right\|_F^2 \right) \right\} .$$

In such a case, it is easy to extend the following ideas to Graph Regularized Non-negative Matrix Factorization. We show here $L$-smad property. We first need the following technical lemma.

**Lemma C.2.** *Let $g_1(\mathbf{U}) = \operatorname{tr}(\mathbf{U}^T \mathcal{L} \mathbf{U})$, then for any $\mathbf{H} \in \mathbb{R}^{M \times K}$ we have $\nabla g_1(\mathbf{U}) = \mathcal{L}\mathbf{U} + \mathcal{L}^T \mathbf{U}$,*

$$\langle \mathbf{H}, \nabla^2 g_1(\mathbf{U})\mathbf{H} \rangle = 2 \langle \mathcal{L}\mathbf{H}, \mathbf{H} \rangle .$$

*Proof.* Note that $\operatorname{tr}(\mathbf{U}^T \mathcal{L} \mathbf{U}) = \langle \mathcal{L}\mathbf{U}, \mathbf{U} \rangle$, now we obtain for $\mathbf{H} \in \mathbb{R}^{M \times K}$ the following

$$\begin{aligned} \langle \mathcal{L}(\mathbf{U} + \mathbf{H}), \mathbf{U} + \mathbf{H} \rangle &= \langle \mathcal{L}(\mathbf{U} + \mathbf{H}), \mathbf{U} + \mathbf{H} \rangle \\ &= \langle \mathcal{L}\mathbf{U}, \mathbf{U} \rangle + \langle \mathcal{L}\mathbf{U}, \mathbf{H} \rangle + \langle \mathcal{L}\mathbf{H}, \mathbf{U} \rangle + \langle \mathcal{L}\mathbf{H}, \mathbf{H} \rangle , \\ &= \langle \mathcal{L}\mathbf{U}, \mathbf{U} \rangle + \langle \mathcal{L}\mathbf{U}, \mathbf{H} \rangle + \langle \mathcal{L}^T \mathbf{U}, \mathbf{H} \rangle + \langle \mathcal{L}\mathbf{H}, \mathbf{H} \rangle . \end{aligned}$$

Thus the statement holds, by collecting the first and second order terms. $\square$

Now, we prove the $L$-smad property.

**Proposition C.3.** *Let* $g(\mathbf{U}, \mathbf{Z}) = \frac{1}{2} \|\mathbf{A} - \mathbf{U}\mathbf{Z}\|_F^2 + \frac{\mu_0}{2} \mathrm{tr}(\mathbf{U}^T \mathcal{L} \mathbf{U})$. *Then, for a certain constant* $L \geq 1$, *the function* $g$ *satisfies L-smad property with respect to the following kernel generating distance,*

$$h_c(\mathbf{U}, \mathbf{Z}) = 3h_1(\mathbf{U}, \mathbf{Z}) + (\|\mathbf{A}\|_F + \mu_0 \|\mathcal{L}\|_F) h_2(\mathbf{U}, \mathbf{Z}).$$

*Proof.* The proof is similar to Proposition 2.1 and Lemma C.2 must be applied for the result. $\square$

Denote $g := \frac{1}{2} \|\mathbf{A} - \mathbf{U}\mathbf{Z}\|_F^2 + \frac{\mu_0}{2} \mathrm{tr}(\mathbf{U}^T \mathcal{L} \mathbf{U})$, $f := \frac{\lambda_0}{2} \left( \|\mathbf{U}\|_F^2 + \|\mathbf{Z}\|_F^2 \right)$ and $h = h_c$.

**Proposition C.4.** *In BPG-MF, with the above defined* $f, g, h$ *the update steps in each iteration are given by* $\mathbf{U^{k+1}} = -r\,\mathbf{P^k}$, $\mathbf{Z^{k+1}} = -r\,\mathbf{Q^k}$ *where* $r \geq 0$ *and satisfies*

$$c_1 \left( \|\mathbf{Q^k}\|_F^2 + \|\mathbf{P^k}\|_F^2 \right) r^3 + (c_2 + \mu_0 \|\mathcal{L}\|_F + \lambda_0)r - 1 = 0, \tag{C.7}$$

*with* $c_1 = 3$ *and* $c_2 = \|\mathbf{A}\|_F$.

The proof is similar to Proposition C.1 and only $c_2$ changes.

## C.3 Extensions to L1-Regularized Matrix Factorization

Now consider the following matrix factorization problem with L1-Regularization

$$\min_{\mathbf{U} \in \mathbb{R}^{M \times K}, \mathbf{Z} \in \mathbb{R}^{K \times N}} \left\{ \Psi(\mathbf{U}, \mathbf{Z}) := \frac{1}{2} \|\mathbf{A} - \mathbf{U}\mathbf{Z}\|_F^2 + \lambda_1 \left( \|\mathbf{U}\|_1 + \|\mathbf{Z}\|_1 \right) \right\}. \tag{C.8}$$

Recall that soft-thresholding operator is defined for any $y \in \mathbb{R}^d$ by

$$\mathcal{S}_\theta(y) = \mathrm{argmin}_{x \in \mathbb{R}^d} \left\{ \theta \|x\|_1 + \frac{1}{2} \|x - y\|^2 \right\} = \max\{|y| - \theta, 0\} \,\mathrm{sgn}(y), \tag{C.9}$$

where $\theta > 0$ and the operations are applied element-wise. We require the following technical result.

**Lemma C.3.** *Let* $\mathbf{Q} \in \mathbb{R}^{A \times B}$ *for some positive integers A and B. Let* $t_0 > 0$ *and let* $t \geq 0$ *then*

$$\min_{\mathbf{X} \in \mathbb{R}^{A \times B}} \left\{ \langle \mathbf{Q}, \mathbf{X} \rangle + t_0 \|\mathbf{X}\|_1 : \|\mathbf{X}\|_F^2 \leq t^2 \right\} = -t \left\| S_{t_0}(-\mathbf{Q}) \right\|_F.$$

*with the minimizer at* $\mathbf{X}^* = t \frac{S_{t_0}(-\mathbf{Q})}{\|S_{t_0}(-\mathbf{Q})\|_F}$ *for* $\|S_{t_0}(-\mathbf{Q})\|_F \neq 0$ *and otherwise all* $\mathbf{X}$ *such that* $\|\mathbf{X}\|_F^2 \leq t^2$ *are minimizers. Moreover we have the following equivalence,*

$$\min_{\mathbf{X} \in \mathbb{R}^{A \times B}} \left\{ \langle \mathbf{Q}, \mathbf{X} \rangle + t_0 \|\mathbf{X}\|_1 : \|\mathbf{X}\|_F^2 \leq t^2 \right\} \equiv \min_{\mathbf{X} \in \mathbb{R}^{A \times B}} \left\{ \langle \mathbf{Q}, \mathbf{X} \rangle + t_0 \|\mathbf{X}\|_1 : \|\mathbf{X}\|_F^2 = t^2 \right\}. \tag{C.10}$$

*Proof.* We have the following equivalence

$$\min_{\mathbf{X} \in \mathbb{R}^{A \times B}} \left\{ \langle \mathbf{Q}, \mathbf{X} \rangle + t_0 \|\mathbf{X}\|_1 : \|\mathbf{X}\|_F^2 \leq t^2 \right\} \equiv -\max_{\mathbf{X} \in \mathbb{R}^{A \times B}} \left\{ \langle -\mathbf{Q}, \mathbf{X} \rangle - t_0 \|\mathbf{X}\|_1 : \|\mathbf{X}\|_F^2 \leq t^2 \right\}.$$

Then the result follows due to [41, Proposition 14] with the minimizer at $\mathbf{X}^* = t \frac{S_{t_0}(-\mathbf{Q})}{\|S_{t_0}(-\mathbf{Q})\|_F}$ for $\|S_{t_0}(-\mathbf{Q})\|_F \neq 0$ and $\mathbf{0}$ otherwise. The equivalence statement in (C.10) follows as $\|\mathbf{X}^*\|_F^2 = t^2$ for $\|S_{t_0}(-\mathbf{Q})\|_F \neq 0$ and otherwise all the points satisfying $\|\mathbf{X}\|_F^2 = t^2$ are minimizers. $\square$

Denote $g := \frac{1}{2} \|\mathbf{A} - \mathbf{U}\mathbf{Z}\|_F^2$, $f := \lambda_1 \left( \|\mathbf{U}\|_1 + \|\mathbf{Z}\|_1 \right)$ and $h = h_a$.

**Proposition C.5.** *In BPG-MF, with the above defined* $g, f, h$ *the update steps in each iteration are given by* $\mathbf{U^{k+1}} = r\mathcal{S}_{\lambda_1 \lambda}(-\mathbf{P^k})$, $\mathbf{Z^{k+1}} = r\mathcal{S}_{\lambda_1 \lambda}(-\mathbf{Q^k})$ *where* $r \geq 0$ *and satisfies*

$$c_1 \left( \left\| \mathcal{S}_{\lambda_1 \lambda} \left( -\mathbf{Q^k} \right) \right\|_F^2 + \left\| \mathcal{S}_{\lambda_1 \lambda} \left( -\mathbf{P^k} \right) \right\|_F^2 \right) r^3 + c_2 r - 1 = 0, \tag{C.11}$$

*with* $c_1 = 3$ *and* $c_2 = \|\mathbf{A}\|_F$.

*Proof.* The proof is similar to that of Proposition C.1, but with certain changes due to the L1 norm in the objective. Consider the following subproblem

$$
(\mathbf{U^{k+1}}, \mathbf{Z^{k+1}}) \in \operatorname*{argmin}_{(\mathbf{U},\mathbf{Z}) \in \mathbb{R}^{M \times K} \times \mathbb{R}^{K \times N}} \left\{ \lambda\lambda_1 \left( \|\mathbf{U}\|_1 + \|\mathbf{Z}\|_1 \right) + \langle \mathbf{P^k}, \mathbf{U} \rangle + \langle \mathbf{Q^k}, \mathbf{Z} \rangle \right.
$$

$$
\left. +c_1 \left( \frac{\|\mathbf{U}\|_F^2 + \|\mathbf{Z}\|_F^2}{2} \right)^2 + c_2 \left( \frac{\|\mathbf{U}\|_F^2 + \|\mathbf{Z}\|_F^2}{2} \right) \right\},
$$

Denote the objective in the above minimization problem as $\mathcal{O}(\mathbf{U^k}, \mathbf{Z^k})$. Now, we show that the following holds

$$
\min_{(\mathbf{U},\mathbf{Z}) \in \mathbb{R}^{M \times K} \times \mathbb{R}^{K \times N}} \left( \mathcal{O}(\mathbf{U^k}, \mathbf{Z^k}) \right)
$$

$$
\equiv \min_{t_1 \geq 0, t_2 \geq 0} \left\{ \min_{(\mathbf{U},\mathbf{Z}) \in \mathbb{R}^{M \times K} \times \mathbb{R}^{K \times N}, \|\mathbf{U}\|_F = t_1, \|\mathbf{Z}\|_F = t_2} \left( \mathcal{O}(\mathbf{U^k}, \mathbf{Z^k}) \right) \right\}, \tag{C.12}
$$

$$
\equiv \min_{t_1 \geq 0, t_2 \geq 0} \left\{ \min_{(\mathbf{U},\mathbf{Z}) \in \mathbb{R}^{M \times K} \times \mathbb{R}^{K \times N}, \|\mathbf{U}\|_F \leq t_1, \|\mathbf{Z}\|_F \leq t_2} \left( \mathcal{O}(\mathbf{U^k}, \mathbf{Z^k}) \right) \right\}. \tag{C.13}
$$

where the first step is a simple rewriting of the objective. The second step is non-trivial. In order to prove (C.13) we rewrite (C.12) as

$$
\min_{t_1 \geq 0, t_2 \geq 0} \left\{ \min_{\mathbf{U_1} \in \mathbb{R}^{M \times K}} \left\{ \langle \mathbf{P^k}, \mathbf{U_1} \rangle + \lambda\lambda_1 \|\mathbf{U}\|_1 : \|\mathbf{U_1}\|_F^2 = t_1 \right\} \right.
$$

$$
+ \min_{\mathbf{Z_1} \in \mathbb{R}^{K \times N}} \left\{ \langle \mathbf{Q^k}, \mathbf{Z_1} \rangle + \lambda\lambda_1 \|\mathbf{Z}\|_1 : \|\mathbf{Z_1}\|_F^2 = t_2 \right\}
$$

$$
\left. +c_1 \left( \frac{t_1^2 + t_2^2}{2} \right)^2 + c_2 \left( \frac{t_1^2 + t_2^2}{2} \right) \right\}.
$$

where the second step (C.13) uses Lemma C.3 and strong convexity of $h$. Now, note the following equivalence due to Lemma C.3

$$
\min_{\mathbf{U_1} \in \mathbb{R}^{M \times K}} \left\{ \langle \mathbf{P^k}, \mathbf{U_1} \rangle + \lambda\lambda_1 \|\mathbf{U}\|_1 : \|\mathbf{U_1}\|_F^2 = t_1 \right\}
$$

$$
\equiv \min_{\mathbf{U_1} \in \mathbb{R}^{M \times K}} \left\{ \langle \mathbf{P^k}, \mathbf{U_1} \rangle + \lambda\lambda_1 \|\mathbf{U}\|_1 : \|\mathbf{U_1}\|_F^2 \leq t_1 \right\}, \tag{C.14}
$$

and

$$
\min_{\mathbf{Z_1} \in \mathbb{R}^{K \times N}} \left\{ \langle \mathbf{Q^k}, \mathbf{Z_1} \rangle + \lambda\lambda_1 \|\mathbf{Z}\|_1 : \|\mathbf{Z_1}\|_F^2 = t_2 \right\}
$$

$$
\equiv \min_{\mathbf{Z_1} \in \mathbb{R}^{K \times N}} \left\{ \langle \mathbf{Q^k}, \mathbf{Z_1} \rangle + \lambda\lambda_1 \|\mathbf{Z}\|_1 : \|\mathbf{Z_1}\|_F^2 \leq t_2 \right\}. \tag{C.15}
$$

We solve the subproblems via the following strategy. Denote

$$
\mathbf{U_1^*}(t_1) \in \operatorname{argmin} \left\{ \langle \mathbf{P^k}, \mathbf{U_1} \rangle + \lambda\lambda_1 \|\mathbf{U}\|_1 : \mathbf{U_1} \in \mathbb{R}^{M \times K}, \|\mathbf{U_1}\|_F^2 \leq t_1 \right\}
$$

$$
\mathbf{Z_1^*}(t_2) \in \operatorname{argmin} \left\{ \langle \mathbf{Q^k}, \mathbf{Z_1} \rangle + \lambda\lambda_1 \|\mathbf{Z}\|_1 : \mathbf{Z_1} \in \mathbb{R}^{K \times N}, \|\mathbf{Z_1}\|_F^2 \leq t_2 \right\}
$$

Then we obtain $(\mathbf{U^{k+1}}, \mathbf{Z^{k+1}}) = (\mathbf{U_1^*}(t_1^*), \mathbf{Z_1^*}(t_2^*))$, where $t_1^*$ and $t_2^*$ are obtained by solving the following two dimensional subproblem

$$
(t_1^*, t_2^*) \in \operatorname*{argmin}_{t_1 \geq 0, t_2 \geq 0} \left\{ \min_{\mathbf{U_1} \in \mathbb{R}^{M \times K}} \left\{ \langle \mathbf{P^k}, \mathbf{U_1} \rangle + \lambda\lambda_1 \|\mathbf{U}\|_1 : \|\mathbf{U_1}\|_F^2 \leq t_1 \right\} \right.
$$

$$
+ \min_{\mathbf{Z_1} \in \mathbb{R}^{K \times N}} \left\{ \langle \mathbf{Q^k}, \mathbf{Z_1} \rangle + \lambda\lambda_1 \|\mathbf{Z}\|_1 : \|\mathbf{Z_1}\|_F^2 \leq t_2 \right\}
$$

$$
\left. +c_1 \left( \frac{t_1 + t_2}{2} \right)^2 + c_2 \left( \frac{t_1 + t_2}{2} \right) \right\}.
$$

Note that inner minimization subproblems can be trivially solved once we obtain $\mathbf{U}_1^*(t_1)$ and $\mathbf{Z}_1^*(t_2)$. Due to Lemma C.3 we obtain the solution to the subproblem in each iteration as follows

$$
\mathbf{U}^{\mathbf{k+1}} = \begin{cases} t_1^* \frac{S_{\lambda\lambda_1}(-\mathbf{P^k})}{\left\| S_{\lambda\lambda_1}(-\mathbf{P^k}) \right\|_F}, & \text{for } \left\| S_{\lambda\lambda_1}(-\mathbf{P^k}) \right\|_F \neq 0, \\ \mathbf{0} & otherwise. \end{cases}
$$

$$
\mathbf{Z}^{\mathbf{k+1}} = \begin{cases} t_2^* \frac{S_{\lambda\lambda_1}(-\mathbf{Q^k})}{\left\| S_{\lambda\lambda_1}(-\mathbf{Q^k}) \right\|_F}, & \text{for } \left\| S_{\lambda\lambda_1}(-\mathbf{Q^k}) \right\|_F \neq 0, \\ \mathbf{0} & otherwise. \end{cases}
$$

We solve for $t_1^*$ and $t_2^*$ with the following two dimensional minimization problem

$$
\underset{t_1 \geq 0, t_2 \geq 0}{\operatorname{argmin}} \left\{ -t_1 \left\| S_{\lambda\lambda_1}(-\mathbf{P^k}) \right\|_F - t_2 \left\| S_{\lambda\lambda_1}(-\mathbf{Q^k}) \right\|_F + c_1 \left( \frac{t_1^2 + t_2^2}{2} \right)^2 + c_2 \left( \frac{t_1^2 + t_2^2}{2} \right) \right\}.
$$

Thus, the solutions $t_1^*$ and $t_2^*$ are the non-negative real roots of the following equations

$$
- \left\| S_{\lambda\lambda_1}(-\mathbf{P^k}) \right\|_F + c_1(t_1^2 + t_2^2)t_1 + c_2 t_1 = 0
$$
$$
- \left\| S_{\lambda\lambda_1}(-\mathbf{Q^k}) \right\|_F + c_1(t_1^2 + t_2^2)t_2 + c_2 t_2 = 0.
$$

Set $t_1 = r \left\| S_{\lambda\lambda_1}(-\mathbf{P^k}) \right\|_F$ and $t_2 = r \left\| S_{\lambda\lambda_1}(-\mathbf{Q^k}) \right\|_F$ for some $r \geq 0$. This results in the following cubic equation,

$$
c_1 \left( \left\| S_{\lambda\lambda_1}(-\mathbf{Q^k}) \right\|_F^2 + \left\| S_{\lambda\lambda_1}(-\mathbf{P^k}) \right\|_F^2 \right) r^3 + c_2 r - 1 = 0,
$$

where the solution is the non-negative real root. $\qquad\square$

### C.4 Extensions with Nuclear Norm Regularization

We start with the notion of Singular Value Shrinkage Operator [14], where given a matrix $\mathbf{Q} \in \mathbb{R}^{A \times B}$ of rank $K$ with Singular Value Decomposition given by $\mathbf{U\Sigma V}^T$ with $\mathbf{U} \in \mathbb{R}^{A \times K}$, $\mathbf{\Sigma} \in \mathbb{R}^{K \times K}$ and $\mathbf{V} \in \mathbb{R}^{K \times N}$ for $t \geq 0$ the output is

$$
\mathcal{D}_t(\mathbf{Q}) = \mathbf{U}\mathcal{S}_t(\mathbf{\Sigma})\mathbf{V}^T, \tag{C.16}
$$

where the soft-thresholding operator is applied only to the singular values. Before we proceed, we require the following technical lemma.

**Lemma C.4.** *Let $\mathbf{Q} \in \mathbb{R}^{A \times B}$ of rank $K$ with Singular Value Decomposition given by $\mathbf{U\Sigma V}^T$ with $\mathbf{U} \in \mathbb{R}^{A \times K}$, $\mathbf{\Sigma} \in \mathbb{R}^{K \times K}$ and $\mathbf{Z} \in \mathbb{R}^{K \times N}$. Let $t \geq 0$ and $\|\mathbf{Q}\|_F \neq 0$ then*

$$
\min_{\mathbf{X} \in \mathbb{R}^{A \times B}} \left\{ \langle \mathbf{Q}, \mathbf{X} \rangle + t_0 \|\mathbf{X}\|_* : \|\mathbf{X}\|_F^2 \leq t^2 \right\} = -t \left\| \mathcal{S}_{t_0}(-\mathbf{\Sigma}) \right\|.
$$

*with $\mathbf{X}^* = t \frac{\mathcal{D}_{t_0}(-\mathbf{Q})}{\|\mathcal{D}_{t_0}(-\mathbf{Q})\|_F}$ if $\|\mathcal{D}_{t_0}(-\mathbf{Q})\| \neq 0$ else any $\mathbf{X}$ such that $\|\mathbf{X}\|_F^2 \leq t^2$ is a minimizer. Moreover we have the following equivalence*

$$
\min_{\mathbf{X} \in \mathbb{R}^{A \times B}} \left\{ \langle \mathbf{Q}, \mathbf{X} \rangle + t_0 \|\mathbf{X}\|_* : \|\mathbf{X}\|_F^2 \leq t^2 \right\} = \min_{\mathbf{X} \in \mathbb{R}^{A \times B}} \left\{ \langle \mathbf{Q}, \mathbf{X} \rangle + t_0 \|\mathbf{X}\|_* : \|\mathbf{X}\|_F^2 = t^2 \right\}. \tag{C.17}
$$

*Proof.* The sub-differential of the nuclear norm [14] is given by

$$
\partial \|\mathbf{X}\|_* = \left\{ \mathbf{UV^T} + \mathbf{W} : \mathbf{W} \in \mathbb{R}^{A \times B}, \mathbf{U^T W} = 0, \mathbf{WV} = 0, \|\mathbf{W}\|_2 \leq 1 \right\}. \tag{C.18}
$$

The normal cone for the set $C_1 = \left\{ \mathbf{X} : \|\mathbf{X}\|_F^2 \leq t^2 \right\}$ is given by

$$
\mathcal{N}_{C_1}(\bar{\mathbf{X}}) = \left\{ \mathbf{V} \in \mathbb{R}^{A \times B} : \left\langle \mathbf{V}, \mathbf{X} - \bar{\mathbf{X}} \right\rangle \leq 0 \text{ for all } \mathbf{X} \in C_1 \right\} \equiv \left\{ \theta\bar{\mathbf{X}} : \theta \geq 0 \right\}.
$$

We consider the following problem

$$
\min_{\mathbf{X} \in \mathbb{R}^{A \times B}} \left\{ \langle \mathbf{Q}, \mathbf{X} \rangle + t_0 \|\mathbf{X}\|_* : \|\mathbf{X}\|_F^2 \leq t^2 \right\}.
$$

and the optimality condition [55, Theorem 10.1, p. 422] results in

$$0 \in \mathbf{Q} + t_0 \partial \|\mathbf{X}\|_* + \mathcal{N}_{C_1}(\mathbf{X}) \,.$$

We follow the strategy from [14, Theorem 2.1]. One can decompose $-\mathbf{Q}$ as

$$-\mathbf{Q} = \mathbf{U_0}\mathbf{\Sigma_0}\mathbf{V_0^T} + \mathbf{U_1}\mathbf{\Sigma_1}\mathbf{V_1^T} \,.$$

where $\mathbf{U_0}, \mathbf{V_0}$ contain the singular vectors for singular values greater than $t_0$ and $\mathbf{U_1}, \mathbf{V_1}$ for less than equal to $t_0$. Then with $\mathbf{X} = \mathbf{U_0}\mathbf{\Sigma}\mathbf{V_0^T}$, the optimality condition becomes

$$0 = \mathbf{Q} + t_0(\mathbf{U_0}\mathbf{V_0^T} + \mathbf{W}) + \theta\mathbf{U_0}\mathbf{\Sigma}\mathbf{V_0^T} \,, \tag{C.19}$$

and thus we obtain

$$\mathbf{U_0}\mathbf{\Sigma_0}\mathbf{V_0^T} + \mathbf{U_1}\mathbf{\Sigma_1}\mathbf{V_1^T} = t_0 \left( \mathbf{U_0}\mathbf{V_0^T} + \mathbf{W} \right) + \theta\mathbf{U_0}\mathbf{\Sigma}\mathbf{V_0^T} \,.$$

With $\mathbf{W} = t_0^{-1}\mathbf{U_1}\mathbf{\Sigma_1}\mathbf{V_1^T}$ all the conditions in (C.18) are satisfied. For some unknown $\theta \geq 0$ we have

$$\theta\mathbf{\Sigma} = \mathbf{\Sigma_0} - t_0\mathbf{I} \,.$$

The objective $\langle \mathbf{Q}, \mathbf{X} \rangle + t_0 \|\mathbf{X}\|_*$ is now monotonically decreasing with $\theta$ after substituting. Thus, we obtain the solution $\mathbf{X} = \frac{t}{\|\mathbf{\Sigma_0} - t_0\mathbf{I}\|}\mathbf{U_0} \left( \mathbf{\Sigma_0} - t_0\mathbf{I} \right) \mathbf{V_0^T}$ for $\|\mathbf{\Sigma_0} - t_0\mathbf{I}\| \neq 0$ else the solution is $\mathbf{0}$. The equivalence statement in (C.17) follows trivially because if $\|\mathbf{\Sigma_0} - t_0\mathbf{I}\| \neq 0$ we have $\|\mathbf{X}\|_F^2 = t^2$ otherwise all the points satisfying $\|\mathbf{X}\|_F^2 \leq t^2$ are minimizers. $\qquad\square$

Here, we want to solve matrix factorization problem with nuclear norm regularization, where for certain constant $\lambda_2 > 0$ we want to solve

$$\min_{\mathbf{U}\in\mathbb{R}^{M\times K}, \mathbf{Z}\in\mathbb{R}^{K\times N}} \left\{ \Psi(\mathbf{U}, \mathbf{Z}) := \frac{1}{2} \|\mathbf{A} - \mathbf{U}\mathbf{Z}\|_F^2 + \lambda_2 \left( \|\mathbf{U}\|_* + \|\mathbf{Z}\|_* \right) \right\} \,. \tag{C.20}$$

Denote $g := \frac{1}{2} \|\mathbf{A} - \mathbf{U}\mathbf{Z}\|_F^2$, $f := \lambda_2 \left( \|\mathbf{U}\|_* + \|\mathbf{Z}\|_* \right)$ and $h = h_a$.

**Proposition C.6.** *In BPG-MF, with the above defined $g, f, h$ the update steps in each iteration are given by* $\mathbf{U^{k+1}} = r\mathcal{D}_{\lambda_1\lambda}(-\mathbf{P^k})$, $\mathbf{Z^{k+1}} = r\mathcal{D}_{\lambda_1\lambda}(-\mathbf{Q^k})$ *where* $r \geq 0$ *and satisfies*

$$c_1 \left( \left\|\mathcal{D}_{\lambda_1\lambda}\left(-\mathbf{Q^k}\right)\right\|_F^2 + \left\|\mathcal{D}_{\lambda_1\lambda}\left(-\mathbf{P^k}\right)\right\|_F^2 \right) r^3 + c_2 r - 1 = 0 \,, \tag{C.21}$$

*with $c_1 = 3$ and $c_2 = \|\mathbf{A}\|_F$.*

The proof is similar to Proposition C.5 but Lemma C.4 must be used instead of Lemma C.3.

## C.5 Extensions with Non-Convex Sparsity Constraints

We want to solve the matrix factorization problem with non-convex sparsity constraints [8]

$$\min_{\mathbf{U}\in\mathbb{R}^{M\times K}, \mathbf{Z}\in\mathbb{R}^{K\times N}} \left\{ \Psi(\mathbf{U}, \mathbf{Z}) := \frac{1}{2} \|\mathbf{A} - \mathbf{U}\mathbf{Z}\|_F^2 : \|\mathbf{U}\|_0 \leq s_1, \|\mathbf{Z}\|_0 \leq s_2, \right\} \,. \tag{C.22}$$

The problem with additional non-negativity constraints, the so called Sparse NMF is considered in Section D.5. Now, denote $g := \frac{1}{2} \|\mathbf{A} - \mathbf{U}\mathbf{Z}\|_F^2$, $f := \mathbf{I}_{\|\mathbf{U}\|_0 \leq s_1} + \mathbf{I}_{\|\mathbf{Z}\|_0 \leq s_2}$ and $h = h_a$. Note that the Assumption C is not valid here, hence CoCaIn BPG-MF theory does not hold and hints at possible extensions of CoCaIn BPG-MF, which is an interesting open question. Before, we proceed, we require the following concept. Let $y \in \mathbb{R}^d$ and without loss of generality we can assume that $|y_1| \geq |y_2| \geq \ldots \geq |y_d|$, then the hard-thresholding operator [41] is given by

$$\mathcal{H}_s(y) = \mathrm{argmin}_{x\in\mathbb{R}^d} \left\{ \|x - y\|^2 : \|x\|_0 \leq s \right\} = \begin{cases} y_i, & i \leq s, \\ 0, & \text{otherwise,} \end{cases} \tag{C.23}$$

where $s > 0$ and the operations are applied element-wise. We require the following technical lemma.

**Lemma C.5.** *Let* $\mathbf{Q} \in \mathbb{R}^{A \times B}$ *for some positive integers $A$ and $B$. Let $t \geq 0$ and $\|\mathbf{Q}\|_F \neq 0$ then*

$$\min_{\mathbf{X} \in \mathbb{R}^{A \times B}} \left\{ \langle \mathbf{Q}, \mathbf{X} \rangle : \|\mathbf{X}\|_F^2 \leq t^2, \|\mathbf{X}\|_0 \leq s \right\} = -t \, \|\mathcal{H}_s(-\mathbf{Q})\| \ .$$

*with the minimizer* $\mathbf{X}^* = \frac{t \mathcal{H}_s(-\mathbf{Q})}{\|\mathcal{H}_s(-\mathbf{Q})\|}$ *if* $\|\mathcal{H}_s(-\mathbf{Q})\| \neq 0$ *else* $\mathbf{X}^* = \mathbf{0}$. *Moreover we have the following equivalence*

$$\min_{\mathbf{X} \in \mathbb{R}^{A \times B}} \left\{ \langle \mathbf{Q}, \mathbf{X} \rangle : \|\mathbf{X}\|_F^2 \leq t^2, \|\mathbf{X}\|_0 \leq s \right\} = \min_{\mathbf{X} \in \mathbb{R}^{A \times B}} \left\{ \langle \mathbf{Q}, \mathbf{X} \rangle : \|\mathbf{X}\|_F^2 = t^2, \|\mathbf{X}\|_0 \leq s \right\} \ .$$

*Proof.* The proof is similar to [41, Proposition 11]. We have

$$\min_{\mathbf{X} \in \mathbb{R}^{A \times B}} \left\{ \langle \mathbf{Q}, \mathbf{X} \rangle : \|\mathbf{X}\|_F^2 \leq t^2, \|\mathbf{X}\|_0 \leq s \right\} = - \max_{\mathbf{X} \in \mathbb{R}^{A \times B}} \left\{ \langle -\mathbf{Q}, \mathbf{X} \rangle : \|\mathbf{X}\|_F^2 \leq t^2, \|\mathbf{X}\|_0 \leq s \right\} ,$$

$$= - \max_{\mathbf{X} \in \mathbb{R}^{A \times B}} \left\{ \langle \mathcal{H}_s(-\mathbf{Q}), \mathbf{X} \rangle : \|\mathbf{X}\|_F^2 \leq t^2 \right\} \ .$$

The first equality is a simple rewriting of the objective. Then, the corresponding objective $\langle -\mathbf{Q}, \mathbf{X} \rangle$ can be maximized with $\sum_{i=1}^A \sum_{j=1}^B \mathbf{I}_{(i,j) \in \Omega_0}(-\mathbf{Q}_{ij} \mathbf{X}_{ij})$ where $\Omega_0$ is set of index pairs and $\mathbf{I}_{(i,j) \in \Omega_0}$ is 1 if the index pair if $(i,j) \in \Omega_0$ and zero otherwise. Note that the objective $\langle -\mathbf{Q}, \mathbf{X} \rangle$ is maximized if $\Omega_0$ contains all the index pairs corresponding to the elements of $-\mathbf{Q}$ with highest absolute value which is captured by Hard-thresholding operator. Thus, the second equality follows and the solution follows due to Lemma C.1. The equivalence statement follows as $\|\mathbf{X}^*\|_F^2 = t^2$ for $\|\mathcal{H}_s(-\mathbf{Q})\| \neq 0$ else the function value is zero and is attained by all the points in the set $\left\{ \mathbf{X} : \|\mathbf{X}\|_F^2 \leq t^2 \right\}$ are minimizers, hence the equivalence. □

**Proposition C.7.** *In BPG-MF, with the above defined $g, f, h$ the update steps in each iteration are given by* $\mathbf{U}^{k+1} = r \mathcal{H}_{s_1}(-\mathbf{P}^k)$, $\mathbf{Z}^{k+1} = r \mathcal{H}_{s_2}(-\mathbf{Q}^k)$ *where $r \geq 0$ and satisfies*

$$c_1 \left( \left\| \mathcal{H}_{s_1}\left(-\mathbf{Q}^k\right) \right\|_F^2 + \left\| \mathcal{H}_{s_2}\left(-\mathbf{P}^k\right) \right\|_F^2 \right) r^3 + c_2 r - 1 = 0 \,, \tag{C.24}$$

*with $c_1 = 3$ and $c_2 = \|\mathbf{A}\|_F$.*

The proof is similar to Proposition C.5 but Lemma C.5 must be used instead of Lemma C.3.

# D  Closed Form Solutions Part II for NMF variants

For simplicity we consider the following problem [36, 37]

$$\min_{\mathbf{U} \in \mathbb{R}^{M \times K}, \mathbf{Z} \in \mathbb{R}^{K \times N}} \left\{ \Psi(\mathbf{U}, \mathbf{Z}) := \frac{1}{2} \|\mathbf{A} - \mathbf{U}\mathbf{Z}\|_F^2 + \mathbf{I}_{\mathbf{U} \geq \mathbf{0}} + \mathbf{I}_{\mathbf{Z} \geq \mathbf{0}} \right\} \ . \tag{D.1}$$

We set $\mathcal{R}_1(\mathbf{U}) = 0$, $\mathcal{R}_2(\mathbf{Z}) = 0$, $g = \Psi$ and $f = \mathbf{I}_{\mathbf{U} \geq \mathbf{0}} + \mathbf{I}_{\mathbf{Z} \geq \mathbf{0}}$ where $\mathbf{I}$ is the indicator operator. We start with the following technical lemma.

**Lemma D.1.** *Let* $\mathbf{Q} \in \mathbb{R}^{A \times B}$ *for some positive integers $A$ and $B$. Let $t \geq 0$ and $\|\mathbf{Q}\|_F \neq 0$ then*

$$\min_{\mathbf{X} \in \mathbb{R}^{A \times B}} \left\{ \langle \mathbf{Q}, \mathbf{X} \rangle : \|\mathbf{X}\|_F^2 \leq t^2, \mathbf{X} \geq \mathbf{0} \right\} = -t \, \|\Pi_+(-\mathbf{Q})\|_F \ ,$$

*with the minimizer* $\mathbf{X}^* = t \frac{\Pi_+(-\mathbf{Q})}{\|\Pi_+(-\mathbf{Q})\|_F}$ *if* $\|\Pi_+(-\mathbf{Q})\|_F \neq 0$ *else* $\mathbf{X}^* = \mathbf{0}$. *For* $\|\Pi_+(-\mathbf{Q})\|_F \neq 0$, *we have the following equivalence*

$$\min_{\mathbf{X} \in \mathbb{R}^{A \times B}} \left\{ \langle \mathbf{Q}, \mathbf{X} \rangle : \|\mathbf{X}\|_F^2 \leq t^2, \mathbf{X} \geq \mathbf{0} \right\} \equiv \min_{\mathbf{X} \in \mathbb{R}^{A \times B}} \left\{ \langle \mathbf{Q}, \mathbf{X} \rangle : \|\mathbf{X}\|_F^2 = t^2, \mathbf{X} \geq \mathbf{0} \right\} \ . \tag{D.2}$$

*Proof.* On rewriting we have the following equivalence

$$\min_{\mathbf{X} \in \mathbb{R}^{A \times B}} \left\{ \langle \mathbf{Q}, \mathbf{X} \rangle : \|\mathbf{X}\|_F^2 \leq t^2, \mathbf{X} \geq \mathbf{0} \right\} \equiv - \max_{\mathbf{X} \in \mathbb{R}^{A \times B}} \left\{ \langle -\mathbf{Q}, \mathbf{X} \rangle : \|\mathbf{X}\|_F^2 \leq t^2, \mathbf{X} \geq \mathbf{0} \right\} \ .$$

The expression $\langle -\mathbf{Q}, \mathbf{X} \rangle$ is maximized at $\mathbf{X}^* = c\Pi_+(-\mathbf{Q})$ for certain constant $c$. On substituting we have

$$\langle -\mathbf{Q}, \mathbf{X}^* \rangle = c \left\| \Pi_+(-\mathbf{Q}) \right\|_F^2 \ .$$

Since, the dependence on $c$ is linear and we additionally require $\|\mathbf{X}\|_F^2 \leq t^2$, we can set $c = \frac{t}{\|\Pi_+(-\mathbf{Q})\|_F}$ if $\|\Pi_+(-\mathbf{Q})\|_F \neq 0$ else $c = 0$. Hence, the minimizer to

$$\min_{\mathbf{X} \in \mathbb{R}^{A \times B}} \left\{ \langle \mathbf{Q}, \mathbf{X} \rangle : \|\mathbf{X}\|_F^2 \leq t^2 \right\}$$

is attained at $\mathbf{X}^* = -t\frac{\Pi_+(-\mathbf{Q})}{\|\Pi_+(-\mathbf{Q})\|_F}$ for $\|\Pi_+(-\mathbf{Q})\|_F \neq 0$ else $\mathbf{X}^* = 0$. The equivalence in the statement follows as $\|\mathbf{X}^*\|_F^2 = t^2$. □

Denote $g = \Psi$, $f = \mathbf{I}_{\mathbf{U} \geq \mathbf{0}} + \mathbf{I}_{\mathbf{Z} \geq \mathbf{0}}$ and $h = h_a$.

**Proposition D.1.** *In BPG-MF, when $g = \Psi$ in* (D.1) *the update step in each iteration are given by* $\mathbf{U}^{\mathbf{k+1}} = \Pi_+(-\mathbf{P^k})$, $\mathbf{Z}^{\mathbf{k+1}} = \Pi_+(-\mathbf{Q^k})$ *where $r \geq 0$ and satisfies*

$$c_1 \left( \left\| \Pi_+(-\mathbf{Q^k}) \right\|_F^2 + \left\| \Pi_+(-\mathbf{P^k}) \right\|_F^2 \right) r^3 + c_2 r - 1 = 0 \,. \tag{D.3}$$

*with $c_1 = 3$ and $c_2 = \|\mathbf{A}\|_F$.*

*Proof.* The proof is similar to that of Proposition C.1, but with certain changes due to the involved non-negativity constraints for the objective. Consider the following subproblem

$$(\mathbf{U}^{\mathbf{k+1}}, \mathbf{Z}^{\mathbf{k+1}}) \in \underset{(\mathbf{U},\mathbf{Z}) \in \mathbb{R}_+^{M \times K} \times \mathbb{R}_+^{K \times N}}{\operatorname{argmin}} \left\{ \langle \mathbf{P^k}, \mathbf{U} \rangle + \langle \mathbf{Q^k}, \mathbf{Z} \rangle \right.$$

$$\left. + c_1 \left( \frac{\|\mathbf{U}\|_F^2 + \|\mathbf{Z}\|_F^2}{2} \right)^2 + c_2 \left( \frac{\|\mathbf{U}\|_F^2 + \|\mathbf{Z}\|_F^2}{2} \right) \right\} \ .$$

Denote the objective in the above minimization problem as $\mathcal{O}(\mathbf{U^k}, \mathbf{Z^k})$. Now, we show that the following holds

$$\min_{(\mathbf{U},\mathbf{Z}) \in \mathbb{R}^{M \times K} \times \mathbb{R}^{K \times N}} \left( \mathcal{O}(\mathbf{U^k}, \mathbf{Z^k}) \right)$$

$$\equiv \min_{t_1 \geq 0, t_2 \geq 0} \left\{ \min_{(\mathbf{U},\mathbf{Z}) \in \mathbb{R}^{M \times K} \times \mathbb{R}^{K \times N}, \|\mathbf{U}\|_F = t_1, \|\mathbf{Z}\|_F = t_2} \left( \mathcal{O}(\mathbf{U^k}, \mathbf{Z^k}) \right) \right\}, \tag{D.4}$$

$$\equiv \min_{t_1 \geq 0, t_2 \geq 0} \left\{ \min_{(\mathbf{U},\mathbf{Z}) \in \mathbb{R}^{M \times K} \times \mathbb{R}^{K \times N}, \|\mathbf{U}\|_F \leq t_1, \|\mathbf{Z}\|_F \leq t_2} \left( \mathcal{O}(\mathbf{U^k}, \mathbf{Z^k}) \right) \right\}, \tag{D.5}$$

where the first step is a simple rewriting of the objective and involved variables and the second equivalence proof is similar to that equivalence of (C.13) and (C.12) in Proposition C.5, which we describe now. The second step is non-trivial. In order to prove (D.5) we rewrite (D.4) as

$$\min_{t_1 \geq 0, t_2 \geq 0} \left\{ \min_{\mathbf{U_1} \in \mathbb{R}^{M \times K}} \left\{ \langle \mathbf{P^k}, \mathbf{U_1} \rangle : \|\mathbf{U_1}\|_F^2 = t_1, \mathbf{U_1} \geq 0 \right\} \right.$$

$$+ \min_{\mathbf{Z_1} \in \mathbb{R}^{K \times N}} \left\{ \langle \mathbf{Q^k}, \mathbf{Z_1} \rangle : \|\mathbf{Z_1}\|_F^2 = t_2, \mathbf{Z_1} \geq 0 \right\}$$

$$\left. + c_1 \left( \frac{t_1^2 + t_2^2}{2} \right)^2 + c_2 \left( \frac{t_1^2 + t_2^2}{2} \right) \right\} \ .$$

where the second step uses Lemma D.1 and strong convexity of $h$. Now, due to Lemma C.3, if $\left\| \Pi_+(-\mathbf{P^k}) \right\|_F \neq 0$ we have

$$\min_{\mathbf{U_1} \in \mathbb{R}^{M \times K}} \left\{ \langle \mathbf{P^k}, \mathbf{U_1} \rangle : \|\mathbf{U_1}\|_F^2 = t_1, \mathbf{U_1} \geq 0 \right\} \equiv \min_{\mathbf{U_1} \in \mathbb{R}^{M \times K}} \left\{ \langle \mathbf{P^k}, \mathbf{U_1} \rangle : \|\mathbf{U_1}\|_F^2 \leq t_1, \mathbf{U_1} \geq 0 \right\},$$

$$\tag{D.6}$$

and similarly if $\left\|\Pi_+(-\mathbf{Q^k})\right\|_F \neq 0$ we have

$$\min_{\mathbf{Z_1}\in\mathbb{R}^{K\times N}}\left\{\langle\mathbf{Q^k},\mathbf{Z_1}\rangle : \|\mathbf{Z_1}\|_F^2 = t_2\,,\mathbf{Z_1}\geq 0\right\} \equiv \min_{\mathbf{Z_1}\in\mathbb{R}^{K\times N}}\left\{\langle\mathbf{Q^k},\mathbf{Z_1}\rangle : \|\mathbf{Z_1}\|_F^2 \leq t_2\,,\mathbf{Z_1}\geq 0\right\}. \tag{D.7}$$

Note that if $\left\|\Pi_+(-\mathbf{P^k})\right\|_F = 0$ and $\left\|\mathbf{P^k}\right\|_F \neq 0$ then the objective

$$\min_{\mathbf{U_1}\in\mathbb{R}^{M\times K}}\left\{\langle\mathbf{P^k},\mathbf{U_1}\rangle : \|\mathbf{U_1}\|_F^2 = t_1\,,\mathbf{U_1}\geq 0\right\}$$

with minimum function value of a positive value $t_1 \min_{i\in[M],\,j\in[K]}\{(\mathbf{P^k})_{i,j}\}$ where we have $[A] = \{1,2,\ldots,A\}$ for a positive integer $A$. Similarly if $\left\|\Pi_+(-\mathbf{Q^k})\right\|_F = 0$ and $\left\|\mathbf{Q^k}\right\|_F \neq 0$ the minimum function value for

$$\min_{\mathbf{Z_1}\in\mathbb{R}^{K\times N}}\left\{\langle\mathbf{Q^k},\mathbf{Z_1}\rangle : \|\mathbf{Z_1}\|_F^2 = t_2\,,\mathbf{Z_1}\geq 0\right\}$$

is a positive value $t_2 \min_{i\in[K],\,j\in[N]}\{(\mathbf{Q^k})_{i,j}\}$. Thus for $\left\|\mathbf{P^k}\right\|_F \neq 0$ with $\left\|\Pi_+(-\mathbf{P^k})\right\|_F = 0$ (or $\left\|\mathbf{Q^k}\right\|_F \neq 0$ with $\left\|\Pi_+(-\mathbf{Q^k})\right\|_F = 0$) the final objective (D.4) is monotonically increasing in $t_1$ (or $t_2$) which will drive $t_1$ (or $t_2$) to 0 due to the constraint $t_1 \geq 0$ (or $t_2 \geq 0$). So, without loss of generality we can consider $\left\|\Pi_+(-\mathbf{Q^k})\right\|_F \neq 0$ and $\left\|\Pi_+(-\mathbf{Q^k})\right\|_F = 0$. Now, we obtain the solutions via the following strategy. Denote

$$\mathbf{U_1^*}(t_1) \in \operatorname{argmin}\left\{\langle\mathbf{P^k},\mathbf{U_1}\rangle : \mathbf{U_1}\in\mathbb{R}_+^{M\times K}\,,\|\mathbf{U_1}\|_F^2 \leq t_1\right\},$$

$$\mathbf{Z_1^*}(t_2) \in \operatorname{argmin}\left\{\langle\mathbf{Q^k},\mathbf{Z_1}\rangle : \mathbf{Z_1}\in\mathbb{R}_+^{K\times N}\,,\|\mathbf{Z_1}\|_F^2 \leq t_2\right\}.$$

Then we obtain $(\mathbf{U^{k+1}},\mathbf{Z^{k+1}}) = (\mathbf{U_1^*}(t_1^*),\mathbf{Z_1^*}(t_2^*))$, where $t_1^*$ and $t_2^*$ are obtained by solving the following two dimensional subproblem

$$\begin{aligned}(t_1^*,t_2^*) \in \operatorname*{argmin}_{t_1\geq 0,t_2\geq 0}\Bigg\{&\min_{\mathbf{U_1}\in\mathbb{R}_+^{M\times K}}\left\{\langle\mathbf{P^k},\mathbf{U_1}\rangle : \|\mathbf{U_1}\|_F^2 \leq t_1\right\}\\&+\min_{\mathbf{Z_1}\in\mathbb{R}_+^{K\times N}}\left\{\langle\mathbf{Q^k},\mathbf{Z_1}\rangle : \|\mathbf{Z_1}\|_F^2 \leq t_2\right\}\\&+c_1\left(\frac{t_1+t_2}{2}\right)^2 + c_2\left(\frac{t_1+t_2}{2}\right)\Bigg\}.\end{aligned}$$

Note that inner minimization subproblems can be trivially solved once we obtain $\mathbf{U_1^*}(t_1)$ and $\mathbf{Z_1^*}(t_2)$. Due to Lemma D.1 we obtain the solution to the subproblem in each iteration as follows

$$\mathbf{U^{k+1}} = \begin{cases} t_1^*\frac{\Pi_+(-\mathbf{P^k})}{\|\Pi_+(-\mathbf{P^k})\|_F}, & \text{for } \left\|\Pi_+(-\mathbf{P^k})\right\|_F \neq 0\,,\\ \mathbf{0}, & otherwise\,.\end{cases}$$

$$\mathbf{Z^{k+1}} = \begin{cases} t_2^*\frac{\Pi_+(-\mathbf{Q^k})}{\|\Pi_+(-\mathbf{Q^k})\|_F}, & \text{for } \left\|\Pi_+(-\mathbf{Q^k})\right\|_F \neq 0\,,\\ \mathbf{0}, & otherwise\,.\end{cases}$$

We solve for $t_1^*$ and $t_2^*$ with the following two dimensional minimization problem

$$\operatorname*{argmin}_{t_1\geq 0,t_2\geq 0}\left\{-t_1\left\|\Pi_+(-\mathbf{P^k})\right\|_F - t_2\left\|\Pi_+(-\mathbf{Q^k})\right\|_F + c_1\left(\frac{t_1^2+t_2^2}{2}\right)^2 + c_2\left(\frac{t_1^2+t_2^2}{2}\right)\right\}.$$

Thus, the solutions $t_1^*$ and $t_2^*$ are the non-negative real roots of the following equations

$$-\left\|\Pi_+(-\mathbf{P^k})\right\|_F + c_1(t_1^2 + t_2^2)t_1 + c_2 t_1 = 0\,,$$
$$-\left\|\Pi_+(-\mathbf{Q^k})\right\|_F + c_1(t_1^2 + t_2^2)t_2 + c_2 t_2 = 0\,.$$

Further simplifications lead to $t_1 = r\left\|\Pi_+(-\mathbf{P^k})\right\|_F$ and $t_2 = r\left\|\Pi_+(-\mathbf{Q^k})\right\|_F$ for some $r \geq 0$. This results in the following cubic equation,

$$c_1\left(\left\|\Pi_+(-\mathbf{Q^k})\right\|_F^2 + \left\|\Pi_+(-\mathbf{P^k})\right\|_F^2\right)r^3 + c_2 r - 1 = 0\,,$$

where the solution is the non-negative real root. $\qquad\square$

## D.1 Extensions to L2-regularized NMF

Here, the goal is solve the following minimization problem

$$\min_{\mathbf{U}\in\mathbb{R}^{M\times K}, \mathbf{Z}\in\mathbb{R}^{K\times N}} \left\{ \Psi(\mathbf{U},\mathbf{Z}) := \frac{1}{2}\|\mathbf{A}-\mathbf{U}\mathbf{Z}\|_F^2 + \frac{\lambda_0}{2}\left(\|\mathbf{U}\|_F^2 + \|\mathbf{Z}\|_F^2\right) + \mathbf{I}_{\mathbf{U}\geq\mathbf{0}} + \mathbf{I}_{\mathbf{Z}\geq\mathbf{0}} \right\}.$$

Denote $g := \frac{1}{2}\|\mathbf{A}-\mathbf{U}\mathbf{Z}\|_F^2 + \frac{\lambda_0}{2}\left(\|\mathbf{U}\|_F^2 + \|\mathbf{Z}\|_F^2\right)$, $f := \mathbf{I}_{\mathbf{U}\geq\mathbf{0}} + \mathbf{I}_{\mathbf{Z}\geq\mathbf{0}}$ and $h = h_b$.

**Proposition D.2.** *In BPG-MF, with above defined $g, f, h$ the update step in each iteration are given by $\mathbf{U}^{\mathbf{k+1}} = \Pi_+(-\mathbf{P^k})$, $\mathbf{Z}^{\mathbf{k+1}} = \Pi_+(-\mathbf{Q^k})$ where $r \geq 0$ and satisfies*

$$c_1\left(\left\|\Pi_+(-\mathbf{Q^k})\right\|_F^2 + \left\|\Pi_+(-\mathbf{P^k})\right\|_F^2\right)r^3 + (c_2+\lambda_0)r - 1 = 0,$$

*with $c_1 = 3$ and $c_2 = \|\mathbf{A}\|_F$.*

The proof is similar to Proposition D.1 with only change in $c_2$.

## D.2 Extensions to L1-regularized NMF

Here, the goal is solve the following minimization problem

$$\min_{\mathbf{U}\in\mathbb{R}^{M\times K}, \mathbf{Z}\in\mathbb{R}^{K\times N}} \left\{ \Psi(\mathbf{U},\mathbf{Z}) := \frac{1}{2}\|\mathbf{A}-\mathbf{U}\mathbf{Z}\|_F^2 + \lambda_1\left(\|\mathbf{U}\|_1 + \|\mathbf{Z}\|_1\right) + \mathbf{I}_{\mathbf{U}\geq\mathbf{0}} + \mathbf{I}_{\mathbf{Z}\geq\mathbf{0}} \right\}.$$

We denote $\mathbf{e_D}$ to be a vector of dimension $\mathbf{D}$ with all its elements set to 1.

**Lemma D.2.** *Let $\mathbf{Q}\in\mathbb{R}^{A\times B}$ for some positive integers $A$ and $B$. Let $t \geq 0$ and $\|\mathbf{Q}\|_F \neq 0$ then*

$$\min_{\mathbf{X}\in\mathbb{R}^{A\times B}} \left\{ \langle\mathbf{Q},\mathbf{X}\rangle + t_0\|\mathbf{X}\|_1 : \|\mathbf{X}\|_F^2 \leq t^2, \mathbf{X}\geq 0 \right\} = -t\left\|\Pi_+(-(\mathbf{Q}+t_0\mathbf{e_A}\mathbf{e_B}^T))\right\|_F$$

*with the minimizer $\mathbf{X}^* = t\frac{\Pi_+(-(\mathbf{Q}+t_0\mathbf{e_A}\mathbf{e_B}^T))}{\|\Pi_+(-(\mathbf{Q}+t_0\mathbf{e_A}\mathbf{e_B}^T))\|_F}$ if the condition $\left\|\Pi_+(-(\mathbf{Q}+t_0\mathbf{e_A}\mathbf{e_B}^T))\right\|_F \neq 0$ holds.*

*Proof.* By using $\mathbf{X}\geq 0$ and the basic trace properties we have the following equivalence

$$\|\mathbf{X}\|_1 = \sum_{i,j}\mathbf{X}_{ij} = \mathbf{e_A}^T\mathbf{X}\mathbf{e_B} = \mathbf{tr}\left(\mathbf{e_A}^T\mathbf{X}\mathbf{e_B}\right) = \mathbf{tr}\left(\mathbf{e_B}\mathbf{e_A}^T\mathbf{X}\right) = \left\langle\mathbf{e_A}\mathbf{e_B}^T,\mathbf{X}\right\rangle,$$

hence we have the following equivalence

$$\min_{\mathbf{X}\in\mathbb{R}^{A\times B}}\left\{\langle\mathbf{Q},\mathbf{X}\rangle + t_0\|\mathbf{X}\|_1 : \|\mathbf{X}\|_F^2 \leq t^2, \mathbf{X}\geq 0\right\}$$

$$\equiv \min_{\mathbf{X}\in\mathbb{R}^{A\times B}}\left\{\langle\mathbf{Q}+t_0\mathbf{e_A}\mathbf{e_B}^T,\mathbf{X}\rangle : \|\mathbf{X}\|_F^2 \leq t^2, \mathbf{X}\geq 0\right\}$$

Now, the solution follows due to Lemma D.1. $\qquad\square$

Denote $g := \frac{1}{2}\|\mathbf{A}-\mathbf{U}\mathbf{Z}\|_F^2$, $f := \lambda_1\left(\|\mathbf{U}\|_1 + \|\mathbf{Z}\|_1\right) + \mathbf{I}_{\mathbf{U}\geq\mathbf{0}} + \mathbf{I}_{\mathbf{Z}\geq\mathbf{0}}$ and $h = h_a$.

**Proposition D.3.** *In BPG-MF, with the above defined $g, f, h$ the update steps in each iteration are given by $\mathbf{U}^{\mathbf{k+1}} = r\Pi_+(-(\mathbf{P^k}+t_0\mathbf{e}_M\mathbf{e}_K^T))$, $\mathbf{Z}^{\mathbf{k+1}} = r\Pi_+(-(\mathbf{Q^k}+t_0\mathbf{e}_K\mathbf{e}_N^T))$ where $r \geq 0$ and satisfies*

$$c_1\left(\left\|\Pi_+(-(\mathbf{P^k}+t_0\mathbf{e}_M\mathbf{e}_K^T))\right\|_F^2 + \left\|\Pi_+(-(\mathbf{Q^k}+t_0\mathbf{e}_K\mathbf{e}_N^T))\right\|_F^2\right)r^3 + c_2 r - 1 = 0,$$

*with $c_1 = 3$, $c_2 = \|\mathbf{A}\|_F$ and $t_0 = \lambda\lambda_1$.*

We skip the proof as it is similar to Proposition D.1.

### D.3 Extensions to Graph Regularized Non-negative Matrix Factorization

Graph Regularized Non-negative Matrix Factorization was proposed in [13]. Here, given $\mathcal{L} \in \mathbb{R}^{M \times M}$ we are interested to solve

$$\min_{\mathbf{U} \in \mathbb{R}^{M \times K}, \mathbf{Z} \in \mathbb{R}^{K \times N}} \left\{ \Psi(\mathbf{U}, \mathbf{Z}) = \frac{1}{2} \|\mathbf{A} - \mathbf{U}\mathbf{Z}\|_F^2 + \frac{\mu_0}{2} \mathbf{tr}(\mathbf{U}^T \mathcal{L} \mathbf{U}) \right.$$
$$\left. + \frac{\lambda_0}{2} \left( \|\mathbf{U}\|_F^2 + \|\mathbf{Z}\|_F^2 \right) + \mathbf{I}_{\mathbf{U} \geq \mathbf{0}} + \mathbf{I}_{\mathbf{Z} \geq \mathbf{0}} \right\} .$$

Recall that

$$h_c(\mathbf{U}, \mathbf{Z}) = 3h_1(\mathbf{U}, \mathbf{Z}) + (\|\mathbf{A}\|_F + \mu_0 \|\mathcal{L}\|_F) h_2(\mathbf{U}, \mathbf{Z}) .$$

Denote $g := \frac{1}{2} \|\mathbf{A} - \mathbf{U}\mathbf{Z}\|_F^2 + \frac{\mu_0}{2} \mathbf{tr}(\mathbf{U}^T \mathcal{L} \mathbf{U})$, $f := \frac{\lambda_0}{2} \left( \|\mathbf{U}\|_F^2 + \|\mathbf{Z}\|_F^2 \right) + \mathbf{I}_{\mathbf{U} \geq \mathbf{0}} + \mathbf{I}_{\mathbf{Z} \geq \mathbf{0}}$ and $h = h_c$.

**Proposition D.4.** *In BPG-MF, with the above defined $f, g, h$ the update steps in each iteration are given by $\mathbf{U}^{\mathbf{k+1}} = r\Pi_+(-\mathbf{P}^{\mathbf{k}})$, $\mathbf{Z}^{\mathbf{k+1}} = r\Pi_+(-\mathbf{Q}^{\mathbf{k}})$ where $r \geq 0$ and satisfies*

$$c_1 \left( \left\|\Pi_+(-\mathbf{Q}^{\mathbf{k}})\right\|_F^2 + \left\|\Pi_+(-\mathbf{P}^{\mathbf{k}})\right\|_F^2 \right) r^3 + (c_2 + \mu_0 \|\mathcal{L}\|_F + \lambda_0)r - 1 = 0, \quad \text{(D.8)}$$

*with $c_1 = 3$ and $c_2 = \|\mathbf{A}\|_F$.*

The proof is similar to Proposition D.1 and only $c_2$ changes.

### D.4 Extensions to Symmetric NMF via Non-Symmetric Relaxation.

In [68], the following optimization problem was proposed in the context of Symmetric NMF where the factors $\mathbf{U}$ and $\mathbf{Z}^T$ are equal. The symmetricity of the factors was lifted via a quadratic penalty terms resulting in the following problem

$$\min_{\mathbf{U} \in \mathbb{R}^{M \times K}, \mathbf{Z} \in \mathbb{R}^{K \times N}} \left\{ \Psi(\mathbf{U}, \mathbf{Z}) := \frac{1}{2} \|\mathbf{A} - \mathbf{U}\mathbf{Z}\|_F^2 + \frac{\lambda_0}{2} \left\|\mathbf{U} - \mathbf{Z}^T\right\|_F^2 + \mathbf{I}_{\mathbf{U} \geq \mathbf{0}} + \mathbf{I}_{\mathbf{Z} \geq \mathbf{0}} \right\} .$$

Now, we prove the $L$-smad property. We need the following technical lemma.

**Lemma D.3.** *Let $g(\mathbf{U}, \mathbf{Z}) = \frac{1}{2} \|\mathbf{A} - \mathbf{U}\mathbf{Z}\|_F^2 + \frac{\lambda_0}{2} \left\|\mathbf{U} - \mathbf{Z}^T\right\|_F^2$ be as defined above, we have the following*

$$\nabla_{\mathbf{U}} g(\mathbf{A}, \mathbf{U}\mathbf{Z}) = \lambda_0 \left( \mathbf{U} - \mathbf{Z}^T \right) - (\mathbf{A} - \mathbf{U}\mathbf{Z})\mathbf{Z}^T$$
$$\nabla_{\mathbf{Z}} g(\mathbf{A}, \mathbf{U}\mathbf{Z}) = \lambda_0 \left( \mathbf{U} - \mathbf{Z}^T \right) + \mathbf{U}^T(\mathbf{A} - \mathbf{U}\mathbf{Z})$$

*and*

$$\langle (\mathbf{H_1}, \mathbf{H_2}), \nabla^2 g(\mathbf{A}, \mathbf{U}\mathbf{Z})(\mathbf{H_1}, \mathbf{H_2}) \rangle$$
$$= -2 \langle \mathbf{A} - \mathbf{U}\mathbf{Z}, \mathbf{H_1}\mathbf{H_2} \rangle + \|\mathbf{U}\mathbf{H_2} + \mathbf{H_1}\mathbf{Z}\|_F^2 + \lambda_0 \left\|\mathbf{H_1} - \mathbf{H_2}^T\right\|_F^2 .$$

*Proof.* The first part of proof for function $\frac{1}{2} \|\mathbf{A} - \mathbf{U}\mathbf{Z}\|_F^2$ follows from Proposition 2.1. For the other term, with the Forbenius dot product, we obtain

$$\frac{\lambda_0}{2} \left\|\mathbf{U} + \mathbf{H_1} - \mathbf{Z}^T - \mathbf{H_2}^T\right\|_F^2$$
$$= \frac{\lambda_0}{2} \left( \left\|\mathbf{U} - \mathbf{Z}^T\right\|_F^2 + 2 \left\langle \mathbf{U} - \mathbf{Z}^T, \mathbf{H_1} - \mathbf{H_2}^T \right\rangle + \left\|\mathbf{H_1} - \mathbf{H_2}^T\right\|_F^2 \right) .$$

Combining with Lemma G.1, the statement follows from the collecting the first order and second order terms. $\qquad \square$

**Proposition D.5.** *Let* $g(\mathbf{U}, \mathbf{Z}) = \frac{1}{2} \|\mathbf{A} - \mathbf{UZ}\|_F^2 + \frac{\lambda_0}{2} \|\mathbf{U} - \mathbf{Z}\|_F^2$. *Then, for a certain constant* $L \geq 1$, *the function* $g$ *satisfies L-smad property with respect to the following kernel generating distance,*

$$h_d(\mathbf{U}, \mathbf{Z}) = 3h_1(\mathbf{U}, \mathbf{Z}) + (\|\mathbf{A}\|_F + 2\lambda_0) \, h_2(\mathbf{U}, \mathbf{Z}) \,.$$

*Proof.* The proof is similar to Proposition 2.1 and Lemma D.3 must be applied for the result. $\qquad\square$

Denote $g := \frac{1}{2} \|\mathbf{A} - \mathbf{UZ}\|_F^2 + \frac{\lambda_0}{2} \|\mathbf{U} - \mathbf{Z}\|_F^2$, $f := \mathbf{I}_{\mathbf{U} \geq 0} + \mathbf{I}_{\mathbf{Z} \geq 0}$ and $h = h_d$.

**Proposition D.6.** *In BPG-MF, with the above defined update steps in each iteration are given by* $\mathbf{U^{k+1}} = r\Pi_+ \left( -\mathbf{P^k} \right)$, $\mathbf{Z^{k+1}} = r\Pi_+ \left( -\mathbf{Q^k} \right)$ *where* $r \geq 0$ *and satisfies*

$$c_1 \left( \left\| \Pi_+ \left( -\mathbf{P^k} \right) \right\|_F^2 + \left\| \Pi_+ \left( -\mathbf{Q^k} \right) \right\|_F^2 \right) r^3 + (c_2 + 2\lambda_0)r - 1 = 0 \,, \tag{D.9}$$

*with* $c_1 = 3$ *and* $c_2 = \|\mathbf{A}\|_F$.

The proof is similar to Proposition D.1 and only $c_2$ changes.

## D.5 Extensions to NMF with Non-Convex Sparsity Constraints (Sparse NMF)

Consider the following problem from [8]

$$\min_{\mathbf{U} \in \mathbb{R}^{M \times K}, \mathbf{Z} \in \mathbb{R}^{K \times N}} \left\{ \Psi(\mathbf{U}, \mathbf{Z}) := \frac{1}{2} \|\mathbf{A} - \mathbf{UZ}\|_F^2 : \mathbf{U} \geq \mathbf{0}, \|\mathbf{U}\|_0 \leq s_1, \mathbf{Z} \geq \mathbf{0}, \|\mathbf{Z}\|_0 \leq s_2, \right\} \,,$$

where $s_1$ and $s_2$ are two known positive integers. Denote $g := \frac{1}{2} \|\mathbf{A} - \mathbf{UZ}\|_F^2$, $f := \mathbf{I}_{\mathbf{U} \geq \mathbf{0}} + \mathbf{I}_{\|\mathbf{U}\|_0 \leq s_1} + \mathbf{I}_{\mathbf{Z} \geq \mathbf{0}} + \mathbf{I}_{\|\mathbf{Z}\|_0 \leq s_2}$ and $h = h_a$. Note that the Assumption C is not valid here, hence CoCaIn BPG-MF theory does not hold and hints at possible extensions of CoCaIn BPG-MF, which is an interesting open question. We start with the following technical lemma.

**Proposition D.7.** *Let* $\mathbf{Q} \in \mathbb{R}^{A \times B}$ *for some positive integers* $A$ *and* $B$. *Let* $t \geq 0$ *and* $\|\mathbf{Q}\|_F \neq 0$ *then*

$$\min_{\mathbf{X} \in \mathbb{R}^{A \times B}} \left\{ \langle \mathbf{Q}, \mathbf{X} \rangle : \|\mathbf{X}\|_F^2 \leq t^2, \|\mathbf{X}\|_0 \leq s, \mathbf{X} \geq \mathbf{0} \right\} = -t \, \|\mathcal{H}_s(\Pi_+(-\mathbf{Q}))\|_F \,.$$

*with the minimizer* $\mathbf{X}^* = t \frac{\mathcal{H}_s(\Pi_+(-\mathbf{Q}))}{\|\mathcal{H}_s(\Pi_+(-\mathbf{Q}))\|_F}$ *if* $\|\mathcal{H}_s(\Pi_+(-\mathbf{Q}))\|_F \neq 0$ *else* $\mathbf{X}^* = \mathbf{0}$. *If* $\|\mathcal{H}_s(\Pi_+(-\mathbf{Q}))\|_F \neq 0$ *we have the following equivalence*

$$\min_{\mathbf{X} \in \mathbb{R}^{A \times B}} \left\{ \langle \mathbf{Q}, \mathbf{X} \rangle : \|\mathbf{X}\|_F^2 \leq t^2, \|\mathbf{X}\|_0 \leq s, \mathbf{X} \geq \mathbf{0} \right\} \tag{D.10}$$

$$\equiv \min_{\mathbf{X} \in \mathbb{R}^{A \times B}} \left\{ \langle \mathbf{Q}, \mathbf{X} \rangle : \|\mathbf{X}\|_F^2 = t^2, \|\mathbf{X}\|_0 \leq s, \mathbf{X} \geq \mathbf{0} \right\} \tag{D.11}$$

*Proof.* We have

$$\min_{\mathbf{X}} \left\{ \langle \mathbf{Q}, \mathbf{X} \rangle : \|\mathbf{X}\|_F^2 \leq t^2, \|\mathbf{X}\|_0 \leq s, \mathbf{X} \geq \mathbf{0} \right\}$$

$$= -\max_{\mathbf{X}} \left\{ \langle -\mathbf{Q}, \mathbf{X} \rangle : \|\mathbf{X}\|_F^2 \leq t^2, \|\mathbf{X}\|_0 \leq s, \mathbf{X} \geq \mathbf{0} \right\} \,,$$

$$= -\max_{\mathbf{X}} \left\{ \langle \Pi_+(-\mathbf{Q}), \mathbf{X} \rangle : \|\mathbf{X}\|_F^2 \leq t^2, \|\mathbf{X}\|_0 \leq s \right\} \,,$$

$$= -\max_{\mathbf{X}} \left\{ \langle \mathcal{H}_s(\Pi_+(-\mathbf{Q})), \mathbf{X} \rangle : \|\mathbf{X}\|_F^2 \leq t^2 \right\} \,.$$

The first equality is a simple rewriting of the objective. Then, the corresponding objective $\langle -\mathbf{Q}, \mathbf{X} \rangle$ can be maximized with $\sum_{i=1}^{A} \sum_{j=1}^{B} \mathbf{I}_{(i,j) \in \Omega_0}(-\mathbf{Q}_{ij}\mathbf{X}_{ij})$ where $\Omega_0$ is set of index pairs and $\mathbf{I}_{(i,j) \in \Omega_0}$ is 1 if the index pair if $(i, j) \in \Omega_0$ and zero otherwise. It is easy to see that the objective $\langle -\mathbf{Q}, \mathbf{X} \rangle$ is maximized if $\Omega_0$ contains all the index pairs corresponding to the elements of $-\mathbf{Q}$ with highest absolute value which is captured by Hard-thresholding operator. However due to the non-negativity constraint if there is any $-\mathbf{Q}_{ij}$ such that it is negative, then since $\mathbf{X}_{ij}$ will be driven to zero. So, before we use the Hard-thresholding operator, we need to use $\Pi_+(.) = \max\{0, .\}$ in second equality. The third equality follows as a consequence of hard sparsity constraint similar to Lemma C.5 and the solution follows due to Lemma C.1. The equivalence statement follows as $\|\mathbf{X}^*\|_F^2 = t^2$. $\qquad\square$

**Proposition D.8.** *In BPG-MF, with the above defined $g, f, h$ the update steps in each iteration are* $\mathbf{U^{k+1}} = r\mathcal{H}_{s_1}(\Pi_+(-\mathbf{P^k}))$, $\mathbf{Z^{k+1}} = r\mathcal{H}_{s_2}(\Pi_+(-\mathbf{Q^k}))$ *where $r \geq 0$ and satisfies*

$$c_1 \left( \left\| \mathcal{H}_{s_1} \left( \Pi_+(-\mathbf{Q^k}) \right) \right\|_F^2 + \left\| \mathcal{H}_{s_2} \left( \Pi_+(-\mathbf{P^k}) \right) \right\|_F^2 \right) r^3 + c_2 r - 1 = 0,$$

*with $c_1 = 3$ and $c_2 = \|\mathbf{A}\|_F$.*

The proof is similar to Proposition D.1.

# E  Matrix Completion Problem

Matrix Completion is an important non-convex optimization problem, which arises in practical real world applications, such as recommender systems [35, 15, 23]. Give a matrix $\mathbf{A}$ where only the values at the index set given by $\Omega$ are given. The goal is obtain the rest of the values. One of the popular strategy is to obtain the factors $\mathbf{U} \in \mathbb{R}^{M \times K}$ and $\mathbf{Z} \in \mathbb{R}^{K \times N}$ for a small positive integer $K$. This is cast into the following problem,

$$\min_{\mathbf{U} \in \mathbb{R}^{M \times K}, \mathbf{Z} \in \mathbb{R}^{K \times N}} \left\{ \Psi(\mathbf{U}, \mathbf{Z}) := \frac{1}{2} \|P_\Omega (\mathbf{A} - \mathbf{UZ})\|_F^2 + \frac{\lambda_0}{2} \left( \|\mathbf{U}\|_F^2 + \|\mathbf{Z}\|_F^2 \right) \right\}, \qquad \text{(E.1)}$$

where $P_\Omega$ is an masking operator over index set $\Omega$ which preserves the given matrix entries and sets others to zero.. We require the following technical lemma.

**Lemma E.1.** *Let $g := \frac{1}{2} \|P_\Omega (\mathbf{A} - \mathbf{UZ})\|_F^2$ be as defined above, we have the following*

$$\nabla_{\mathbf{U}} g(\mathbf{A}, \mathbf{UZ}) = -P_\Omega(\mathbf{A} - \mathbf{UZ})\mathbf{Z}^T, \quad \nabla_{\mathbf{Z}} g(\mathbf{A}, \mathbf{UZ}) = -\mathbf{U}^T P_\Omega(\mathbf{A} - \mathbf{UZ})$$

$$\langle (\mathbf{H_1}, \mathbf{H_2}), \nabla^2 g(\mathbf{A}, \mathbf{UZ})(\mathbf{H_1}, \mathbf{H_2}) \rangle = \|P_\Omega(\mathbf{UH_2} + \mathbf{H_1Z})\|_F^2 - 2 \langle P_\Omega(\mathbf{A} - \mathbf{UZ}), \mathbf{H_1H_2} \rangle.$$

*Proof.* With the Forbenius dot product, we have

$$\|P_\Omega(\mathbf{A} - \mathbf{UZ})\|_F^2 = \langle P_\Omega(\mathbf{A} - \mathbf{UZ}), P_\Omega(\mathbf{A} - \mathbf{UZ}) \rangle.$$

In the above expression by substituting $\mathbf{U}$ with $\mathbf{U} + \mathbf{H_1}$ and $\mathbf{Z}$ with $\mathbf{Z} + \mathbf{H_2}$, we obtain

$$\langle P_\Omega(\mathbf{A} - (\mathbf{U} + \mathbf{H_1})(\mathbf{Z} + \mathbf{H_2})), P_\Omega(\mathbf{A} - (\mathbf{U} + \mathbf{H_1})(\mathbf{Z} + \mathbf{H_2})) \rangle,$$
$$= \|P_\Omega(\mathbf{A} - \mathbf{UZ})\|_F^2 + \|P_\Omega(\mathbf{UH_2} + \mathbf{H_1Z})\|_F^2$$
$$- 2 \langle P_\Omega(\mathbf{A} - \mathbf{UZ}), P_\Omega(\mathbf{UH_2} + \mathbf{H_1Z}) \rangle - 2 \langle P_\Omega(\mathbf{A} - \mathbf{UZ}), P_\Omega(\mathbf{H_1H_2}) \rangle$$

where in the last term we ignored the terms higher than second order. Collecting all the first order terms we have

$$- 2 \langle P_\Omega(\mathbf{A} - \mathbf{UZ}), P_\Omega(\mathbf{UH_2} + \mathbf{H_1Z}) \rangle$$
$$= -2 \langle P_\Omega(\mathbf{A} - \mathbf{UZ}), \mathbf{UH_2} + \mathbf{H_1Z} \rangle$$
$$= -2 \langle P_\Omega(\mathbf{A} - \mathbf{UZ})\mathbf{Z}^T, \mathbf{H_1} \rangle - 2 \langle \mathbf{U}^T P_\Omega(\mathbf{A} - \mathbf{UZ}), \mathbf{H_2} \rangle$$

and similarly collecting all the second order terms we have

$$\|P_\Omega(\mathbf{UH_2} + \mathbf{H_1Z})\|_F^2 - 2 \langle P_\Omega(\mathbf{A} - \mathbf{UZ}), P_\Omega(\mathbf{H_1H_2}) \rangle$$
$$= \|P_\Omega(\mathbf{UH_2} + \mathbf{H_1Z})\|_F^2 - 2 \langle P_\Omega(\mathbf{A} - \mathbf{UZ}), \mathbf{H_1H_2} \rangle$$

Thus the statement follows using the second order Taylor expansion. $\qquad \square$

**Proposition E.1.** *Let $g := \frac{1}{2} \|P_\Omega (\mathbf{A} - \mathbf{UZ})\|_F^2$ and $h_1, h_2$ be as defined as in (2.6). Then, for a certain constant $L \geq 1$, the function $g$ satisfies $L$-smad property with respect to the following kernel generating distance,*

$$h_a(\mathbf{U}, \mathbf{Z}) = 3h_1(\mathbf{U}, \mathbf{Z}) + \|P_\Omega(\mathbf{A})\|_F \, h_2(\mathbf{U}, \mathbf{Z}).$$

*Proof.* With Lemma G.1 we obtain

$$
\begin{aligned}
&\left\langle (\mathbf{H}_1, \mathbf{H}_2), \nabla^2 g(A, \mathbf{UZ})(\mathbf{H}_1, \mathbf{H}_2) \right\rangle \\
&= \left\| P_\Omega(\mathbf{UH_2} + \mathbf{H_1 Z}) \right\|_F^2 - 2 \left\langle P_\Omega(\mathbf{A} - \mathbf{UZ}), \mathbf{H_1 H_2} \right\rangle \\
&\leq \left\| \mathbf{H_1 Z} + \mathbf{UH_2} \right\|_F^2 - 2 \left\langle P_\Omega(\mathbf{A} - \mathbf{UZ}), \mathbf{H_1 H_2} \right\rangle \\
&\leq 2 \left\| \mathbf{H_1 Z} \right\|_F^2 + 2 \left\| \mathbf{UH_2} \right\|_F^2 + 2 \left\| P_\Omega(\mathbf{A}) \right\|_F \left\| \mathbf{H_1 H_2} \right\|_F + 2 \left\| P_\Omega(\mathbf{UZ}) \right\|_F \left\| \mathbf{H_1 H_2} \right\|_F \,, \\
&\leq 2 \left\| \mathbf{H_1 Z} \right\|_F^2 + 2 \left\| \mathbf{UH_2} \right\|_F^2 + 2 \left\| P_\Omega(\mathbf{A}) \right\|_F \left\| \mathbf{H_1 H_2} \right\|_F + 2 \left\| \mathbf{UZ} \right\|_F \left\| \mathbf{H_1 H_2} \right\|_F \,.
\end{aligned}
$$

The rest of the proof is similar to Proposition 2.1. $\qquad\square$

**Proposition E.2.** *Let* $g := \frac{1}{2} \left\| P_\Omega \left( \mathbf{A} - \mathbf{UZ} \right) \right\|_F^2 + \frac{\lambda_0}{2} \left( \left\| \mathbf{U} \right\|_F^2 + \left\| \mathbf{Z} \right\|_F^2 \right)$ *and* $h_1, h_2$ *be as defined as in (2.6). Then, for a certain constant* $L \geq 1$*, the function* $g$ *satisfies L-smad property with respect to the following kernel generating distance,*

$$
h_a(\mathbf{U}, \mathbf{Z}) = 3 h_1(\mathbf{U}, \mathbf{Z}) + \left( \left\| P_\Omega(\mathbf{A}) \right\|_F + \lambda_0 \right) h_2(\mathbf{U}, \mathbf{Z}) \,.
$$

The update steps are very similar as what we described earlier in Section C and D.

# F  Closed Form Solution with 5th-order Polynomial

The goal of this section is to show a case, where while obtaining the update step of BPG-MF we obtain a 5th order polynomial equation, for which Newton based method solvers can be used. We later show that we can obtain a cubic equation by slightly modifying the kernel generating distance. Let $\lambda_0 > 0$ and we consider the following problem

$$
\min_{\mathbf{U} \in \mathbb{R}^{M \times K}, \mathbf{Z} \in \mathbb{R}^{K \times N}} \left\{ \Psi(\mathbf{U}, \mathbf{Z}) := \frac{1}{2} \left\| \mathbf{A} - \mathbf{UZ} \right\|_F^2 + \frac{\lambda_0}{2} \left\| \mathbf{U} \right\|_F^2 \right\} . \tag{F.1}
$$

We set $\mathcal{R}_1(\mathbf{U}) = \frac{\lambda_0}{2} \left\| \mathbf{U} \right\|_F^2$, $\mathcal{R}_2(\mathbf{Z}) = 0$, $g = \frac{1}{2} \left\| \mathbf{A} - \mathbf{UZ} \right\|_F^2$, $f(\mathbf{U}, \mathbf{Z}) = \frac{\lambda_0}{2} \left\| \mathbf{U} \right\|_F^2$ and $h = h_a$.

**Proposition F.1.** *In BPG-MF, with above defined* $g, f, h$ *the update steps in each iteration are given by* $\mathbf{U^{k+1}} = -\frac{\mathbf{P^k}}{r_1 + \lambda_0}$, $\mathbf{Z^{k+1}} = -\frac{\mathbf{Q^k}}{r_1}$ *where* $r_1 \geq 0$ *and satisfies*

$$
c_1 \left( \left\| \mathbf{Q^k} \right\|_F^2 (r_1 + \lambda_0)^2 + \left\| \mathbf{P^k} \right\|_F^2 r_1^2 \right) + c_2 r_1^2 (r_1 + \lambda_0)^2 - r_1^3 (r_1 + \lambda_0)^2 = 0 \,, \tag{F.2}
$$

*with* $c_1 = 3$ *and* $c_2 = \left\| \mathbf{A} \right\|_F$.

*Proof.* The proof is similar to that of Proposition C.1. Consider the following subproblem

$$
\begin{aligned}
(\mathbf{U^{k+1}}, \mathbf{Z^{k+1}}) \in \operatorname*{argmin}_{(\mathbf{U}, \mathbf{Z}) \in \mathbb{R}^{M \times K} \times \mathbb{R}^{K \times N}} &\left\{ \frac{\lambda_0}{2} \left\| \mathbf{U} \right\|_F^2 + \left\langle \mathbf{P^k}, \mathbf{U} \right\rangle + \left\langle \mathbf{Q^k}, \mathbf{Z} \right\rangle \right. \\
&\left. + c_1 \left( \frac{\left\| \mathbf{U} \right\|_F^2 + \left\| \mathbf{Z} \right\|_F^2}{2} \right)^2 + c_2 \left( \frac{\left\| \mathbf{U} \right\|_F^2 + \left\| \mathbf{Z} \right\|_F^2}{2} \right) \right\} \,,
\end{aligned}
$$

Denote the objective in the above minimization problem as $\mathcal{O}(\mathbf{U^k}, \mathbf{Z^k})$. Now, we show that the following holds

$$
\begin{aligned}
&\min_{(\mathbf{U}, \mathbf{Z}) \in \mathbb{R}^{M \times K} \times \mathbb{R}^{K \times N}} \left( \mathcal{O}(\mathbf{U^k}, \mathbf{Z^k}) \right) \\
&\equiv \min_{t_1 \geq 0, t_2 \geq 0} \left\{ \min_{(\mathbf{U}, \mathbf{Z}) \in \mathbb{R}^{M \times K} \times \mathbb{R}^{K \times N}, \left\| \mathbf{U} \right\|_F = t_1, \left\| \mathbf{Z} \right\|_F = t_2} \left( \mathcal{O}(\mathbf{U^k}, \mathbf{Z^k}) \right) \right\} \,, \\
&\equiv \min_{t_1 \geq 0, t_2 \geq 0} \left\{ \min_{(\mathbf{U}, \mathbf{Z}) \in \mathbb{R}^{M \times K} \times \mathbb{R}^{K \times N}, \left\| \mathbf{U} \right\|_F \leq t_1, \left\| \mathbf{Z} \right\|_F \leq t_2} \left( \mathcal{O}(\mathbf{U^k}, \mathbf{Z^k}) \right) \right\} \,.
\end{aligned}
$$

where the first step is a simple rewriting of the objective and the second step follows as there is no change in the constraint set and due to Lemma C.1, which is given precisely in Proposition C.1 where

the equivalence argument used for (C.4) and (C.3) holds here. Note that in the first step, we used $\|\mathbf{U}\|_F = t_1$ this results in deviation of value of $c_2$ to $c_2 + \lambda_0$, corresponding to $\mathbf{U}$ (see below). We solve for $(\mathbf{U^{k+1}}, \mathbf{Z^{k+1}})$ via the following strategy. Denote

$$\mathbf{U}_1^*(t_1) \in \operatorname{argmin}\left\{\langle \mathbf{P^k}, \mathbf{U_1}\rangle : \mathbf{U_1} \in \mathbb{R}^{M\times K}, \|\mathbf{U_1}\|_F^2 \le t_1\right\},$$

$$\mathbf{Z}_1^*(t_2) \in \operatorname{argmin}\left\{\langle \mathbf{Q^k}, \mathbf{Z_1}\rangle : \mathbf{Z_1} \in \mathbb{R}^{K\times N}, \|\mathbf{Z_1}\|_F^2 \le t_2\right\}.$$

Then we obtain $(\mathbf{U^{k+1}}, \mathbf{Z^{k+1}}) = (\mathbf{U}_1^*(t_1^*), \mathbf{Z}_1^*(t_2^*))$, where $t_1^*$ and $t_2^*$ are obtained by solving the following two dimensional subproblem

$$(t_1^*, t_2^*) \in \operatorname*{argmin}_{t_1\ge 0, t_2 \ge 0}\left\{\min_{\mathbf{U_1}\in\mathbb{R}^{M\times K}}\left\{\langle\mathbf{P^k}, \mathbf{U_1}\rangle : \|\mathbf{U_1}\|_F^2 \le t_1\right\}\right.$$
$$+ \min_{\mathbf{Z_1}\in\mathbb{R}^{K\times N}}\left\{\langle\mathbf{Q^k}, \mathbf{Z_1}\rangle : \|\mathbf{Z_1}\|_F^2 \le t_2\right\}$$
$$\left. + c_1\left(\frac{t_1^2 + t_2^2}{2}\right)^2 + c_2\frac{t_2^2}{2} + (c_2 + \lambda_0)\frac{t_1^2}{2}\right\}.$$

Note that inner minimization subproblems can be trivially solved once we obtain $\mathbf{U}_1^*(t_1)$ and $\mathbf{Z}_1^*(t_2)$ via Lemma C.1. Then the solution to the subproblem in each iteration as follows:

$$\mathbf{U^{k+1}} = \begin{cases} t_1^* \frac{-\mathbf{P^k}}{\|\mathbf{P^k}\|_F}, & \text{for } \|\mathbf{P^k}\|_F \ne 0, \\ \mathbf{0} & otherwise. \end{cases}$$

$$\mathbf{Z^{k+1}} = \begin{cases} t_2^* \frac{-\mathbf{Q^k}}{\|\mathbf{Q^k}\|_F}, & \text{for } \|\mathbf{Q^k}\|_F \ne 0, \\ \mathbf{0} & otherwise. \end{cases}$$

We solve for $t_1^*$ and $t_2^*$ with the following two dimensional minimization problem

$$\operatorname*{argmin}_{t_1\ge 0, t_2\ge 0}\left\{-t_1\|\mathbf{P^k}\|_F - t_2\|\mathbf{Q^k}\|_F + c_1\left(\frac{t_1^2 + t_2^2}{2}\right)^2 + c_2\frac{t_2^2}{2} + (c_2+\lambda_0)\frac{t_1^2}{2}\right\}.$$

Thus, the solutions $t_1^*$ and $t_2^*$ are the non-negative real roots of the following equations

$$-\|\mathbf{P^k}\|_F + c_1(t_1^2 + t_2^2)t_1 + (c_2 + \lambda_0)t_1 = 0, \tag{F.3}$$

$$-\|\mathbf{Q^k}\|_F + c_1(t_1^2 + t_2^2)t_2 + c_2 t_2 = 0. \tag{F.4}$$

Further simplifications with $t_1 = \frac{\|\mathbf{P^k}\|_F}{r_1+\lambda_0}$ and $t_2 = \frac{\|\mathbf{Q^k}\|_F}{r_1}$ denoting $r_1 = c_1(t_1^2 + t_2^2) + c_2$, then we have

$$r_1 = c_1\left(\left(\frac{\|\mathbf{P^k}\|_F}{r_1 + \lambda_0}\right)^2 + \left(\frac{\|\mathbf{Q^k}\|_F}{r_1}\right)^2\right) + c_2$$

This will result in following $5^{th}$ order equation,

$$c_1\left(\|\mathbf{P^k}\|_F^2 r_1^2 + \|\mathbf{Q^k}\|_F^2 (r_1 + \lambda_0)^2\right) + c_2 r_1^2(r_1 + \lambda_0)^2 - r_1^3(r_1 + \lambda_0)^2 = 0.$$

$\square$

## F.1 Conversion to Cubic Equation

We set $\mathcal{R}_1(\mathbf{U}) = \frac{\lambda_0}{2}\|\mathbf{U}\|_F^2$, $\mathcal{R}_2(\mathbf{Z}) = 0$ and $g = \frac{1}{2}\|\mathbf{A} - \mathbf{UZ}\|_F^2$. Denote $f(\mathbf{U}, \mathbf{Z}) = \frac{\lambda_0}{2}\|\mathbf{U}\|_F^2$, $h(\mathbf{U}, \mathbf{Z}) = h_a(\mathbf{U}, \mathbf{Z}) + \frac{\lambda_0}{2}\|\mathbf{Z}\|_F^2$. Note that such a $g$ satisfies $L$-smad property with respect to $h$ satisfies $L$-smad trivially since only a quadratic term is added to $h_a$.

**Proposition F.2.** *In BPG-MF, with the above defined $g, f, h$ the update steps in each iteration are given by* $\mathbf{U^{k+1}} = -r\,\mathbf{P^k}$, $\mathbf{Z^{k+1}} = -r\,\mathbf{Q^k}$ *where $r$ is the non-negative real root of*

$$c_1\left(\|\mathbf{Q^k}\|_F^2 + \|\mathbf{P^k}\|_F^2\right)r^3 + (c_2 + \lambda_0)r - 1 = 0, \tag{F.5}$$

*with $c_1 = 3$ and $c_2 = \|\mathbf{A}\|_F$.*

*Proof.* The resulting subproblem is

$$(\mathbf{U^{k+1}}, \mathbf{Z^{k+1}}) \in \underset{(\mathbf{U},\mathbf{Z})\in\mathbb{R}^{M\times K}\times\mathbb{R}^{K\times N}}{\operatorname{argmin}} \left\{ \langle \mathbf{P^k}, \mathbf{U} \rangle + \langle \mathbf{Q^k}, \mathbf{Z} \rangle \right.$$

$$\left. +c_1 \left( \frac{\|\mathbf{U}\|_F^2 + \|\mathbf{Z}\|_F^2}{2} \right)^2 + (c_2 + \lambda_0) \left( \frac{\|\mathbf{U}\|_F^2 + \|\mathbf{Z}\|_F^2}{2} \right) \right\}.$$

The rest of the proof is similar to Proposition C.1. $\qquad\square$

### F.2 Extensions to Mixed Regularization Terms

Let $\lambda_0 > 0$ and we consider the following problem

$$\min_{\mathbf{U}\in\mathbb{R}^{M\times K}, \mathbf{Z}\in\mathbb{R}^{K\times N}} \left\{ \Psi(\mathbf{U}, \mathbf{Z}) := \frac{1}{2} \|\mathbf{A} - \mathbf{UZ}\|_F^2 + \frac{\lambda_0}{2} \|\mathbf{U}\|_F^2 + \lambda_1 \|\mathbf{Z}\|_1 \right\}. \tag{F.6}$$

Note that the regularizer is a mixture of L1 and L2 regularization. The usual strategy with $h = h_a$ would result in a fifth order polynomial. In order to generate a cubic equation, we use the same strategy as given Section F.1. We set $h(\mathbf{U}, \mathbf{Z}) = h_a(\mathbf{U}, \mathbf{Z}) + \frac{\lambda_0}{2} \|\mathbf{Z}\|_F^2$, $g = \frac{1}{2} \|\mathbf{A} - \mathbf{UZ}\|_F^2$ and $f(\mathbf{U}, \mathbf{Z}) = \frac{\lambda_0}{2} \|\mathbf{U}\|_F^2 + \lambda_1 \|\mathbf{Z}\|_1$.

**Proposition F.3.** *In BPG-MF, with the above defined $g, f, h$ the update steps in each iteration are given by $\mathbf{U^{k+1}} = -r\, \mathbf{P^k}$, $\mathbf{Z^{k+1}} = r\mathcal{S}_{\lambda\lambda_1} \left( -\mathbf{Q^k} \right)$ where $r$ is the non-negative real root of*

$$c_1 \left( \|\mathbf{P^k}\|_F^2 + \|\mathcal{S}_{\lambda\lambda_1} \left( -\mathbf{Q^k} \right)\|_F^2 \right) r^3 + (c_2 + \lambda_0)r - 1 = 0, \tag{F.7}$$

*with $c_1 = 3$ and $c_2 = \|\mathbf{A}\|_F$.*

The proof is similar to Proposition C.1 and Proposition C.5.

## G Technical Lemmas and Proofs

Before we proceed to the proof of Proposition 2.1 we require the following technical lemma.

**Lemma G.1.** *Let $g := \frac{1}{2} \|\mathbf{A} - \mathbf{UZ}\|_F^2$, then we have the following*

$$\nabla g(\mathbf{A}, \mathbf{UZ}) = \left( -(\mathbf{A} - \mathbf{UZ})\mathbf{Z}^T, -\mathbf{U}^T(\mathbf{A} - \mathbf{UZ}) \right)$$

$$\langle (\mathbf{H_1}, \mathbf{H_2}), \nabla^2 g(\mathbf{A}, \mathbf{UZ})(\mathbf{H_1}, \mathbf{H_2}) \rangle = -2 \langle \mathbf{A} - \mathbf{UZ}, \mathbf{H_1H_2} \rangle + \langle \mathbf{UH_2} + \mathbf{H_1Z}, \mathbf{UH_2} + \mathbf{H_1Z} \rangle.$$

*Proof.* With the Forbenius dot product, we have

$$\|\mathbf{A} - \mathbf{UZ}\|_F^2 = \langle \mathbf{A} - \mathbf{UZ}, \mathbf{A} - \mathbf{UZ} \rangle.$$

In the above expression by substituting $\mathbf{U}$ with $\mathbf{U} + \mathbf{H_1}$ and $\mathbf{Z}$ with $\mathbf{Z} + \mathbf{H_2}$, we obtain

$$\langle \mathbf{A} - (\mathbf{U} + \mathbf{H_1})(\mathbf{Z} + \mathbf{H_2}), \mathbf{A} - (\mathbf{U} + \mathbf{H_1})(\mathbf{Z} + \mathbf{H_2}) \rangle,$$

$$= \langle \mathbf{A} - \mathbf{UZ} - \mathbf{UH_2} - \mathbf{H_1Z} - \mathbf{H_1H_2}, \mathbf{A} - \mathbf{UZ} - \mathbf{UH_2} - \mathbf{H_1Z} - \mathbf{H_1H_2} \rangle,$$

$$= \langle \mathbf{A}, \mathbf{A} \rangle - \langle \mathbf{A}, \mathbf{UZ} \rangle - \langle \mathbf{A}, \mathbf{UH_2} \rangle - \langle \mathbf{A}, \mathbf{H_1Z} \rangle - \langle \mathbf{A}, \mathbf{H_1H_2} \rangle,$$

$$- \langle \mathbf{UZ}, \mathbf{A} \rangle + \langle \mathbf{UZ}, \mathbf{UZ} \rangle + \langle \mathbf{UZ}, \mathbf{UH_2} \rangle + \langle \mathbf{UZ}, \mathbf{H_1Z} \rangle + \langle \mathbf{UZ}, \mathbf{H_1H_2} \rangle$$

$$- \langle \mathbf{UH_2}, \mathbf{A} \rangle + \langle \mathbf{UH_2}, \mathbf{UZ} \rangle + \langle \mathbf{UH_2}, \mathbf{UH_2} \rangle + \langle \mathbf{UH_2}, \mathbf{H_1Z} \rangle + \langle \mathbf{UH_2}, \mathbf{H_1H_2} \rangle$$

$$- \langle \mathbf{H_1Z}, \mathbf{A} \rangle + \langle \mathbf{H_1Z}, \mathbf{UZ} \rangle + \langle \mathbf{H_1Z}, \mathbf{UH_2} \rangle + \langle \mathbf{H_1Z}, \mathbf{H_1Z} \rangle + \langle \mathbf{H_1Z}, \mathbf{H_1H_2} \rangle$$

$$- \langle \mathbf{H_1H_2}, \mathbf{A} \rangle + \langle \mathbf{H_1H_2}, \mathbf{UZ} \rangle + \langle \mathbf{H_1H_2}, \mathbf{UH_2} \rangle + \langle \mathbf{H_1H_2}, \mathbf{H_1Z} \rangle + \langle \mathbf{H_1H_2}, \mathbf{H_1H_2} \rangle.$$

Collecting all the first order terms we have

$$- \langle \mathbf{A}, \mathbf{UH_2} \rangle - \langle \mathbf{A}, \mathbf{H_1Z} \rangle + \langle \mathbf{UZ}, \mathbf{UH_2} \rangle + \langle \mathbf{UZ}, \mathbf{H_1Z} \rangle$$

$$- \langle \mathbf{UH_2}, \mathbf{A} \rangle + \langle \mathbf{UH_2}, \mathbf{UZ} \rangle - \langle \mathbf{H_1Z}, \mathbf{A} \rangle + \langle \mathbf{H_1Z}, \mathbf{UZ} \rangle$$

$$= - \langle \mathbf{A}, \mathbf{H_1Z} \rangle + \langle \mathbf{UZ}, \mathbf{H_1Z} \rangle - \langle \mathbf{H_1Z}, \mathbf{A} \rangle + \langle \mathbf{H_1Z}, \mathbf{UZ} \rangle$$

$$- \langle \mathbf{A}, \mathbf{UH_2} \rangle + \langle \mathbf{UZ}, \mathbf{UH_2} \rangle - \langle \mathbf{UH_2}, \mathbf{A} \rangle + \langle \mathbf{UH_2}, \mathbf{UZ} \rangle,$$

$$= -2 \langle \mathbf{A}, \mathbf{H_1Z} \rangle - 2 \langle \mathbf{A}, \mathbf{UH_2} \rangle + 2 \langle \mathbf{UZ}, \mathbf{H_1Z} \rangle + 2 \langle \mathbf{UZ}, \mathbf{UH_2} \rangle,$$

$$= -2\mathbf{tr}((\mathbf{A} - \mathbf{UZ})\mathbf{Z}^T\mathbf{H_1}^T) - 2\mathbf{tr}((\mathbf{A} - \mathbf{UZ})\mathbf{H_2}^T\mathbf{U}^T),$$

$$= -2\mathbf{tr}((\mathbf{A} - \mathbf{UZ})\mathbf{Z}^T\mathbf{H_1}^T) - 2\mathbf{tr}(\mathbf{U}^T(\mathbf{A} - \mathbf{UZ})\mathbf{H_2}^T),$$

and similarly collecting all the second order terms we have

$$-\langle \mathbf{A}, \mathbf{H_1 H_2}\rangle + \langle \mathbf{UZ}, \mathbf{H_1 H_2}\rangle + \langle \mathbf{UH_2}, \mathbf{UH_2}\rangle + \langle \mathbf{UH_2}, \mathbf{H_1 Z}\rangle$$
$$+ \langle \mathbf{H_1 Z}, \mathbf{UH_2}\rangle + \langle \mathbf{H_1 Z}, \mathbf{H_1 Z}\rangle - \langle \mathbf{H_1 H_2}, \mathbf{A}\rangle + \langle \mathbf{H_1 H_2}, \mathbf{UZ}\rangle$$
$$= -2\langle \mathbf{A} - \mathbf{UZ}, \mathbf{H_1 H_2}\rangle + \langle \mathbf{UH_2} + \mathbf{H_1 Z}, \mathbf{UH_2} + \mathbf{H_1 Z}\rangle .$$

Thus the statement follows using the second order Taylor expansion. $\qquad\square$

**Lemma G.2.** *Given* $h_1 := \left(\frac{\|\mathbf{U}\|_F^2 + \|\mathbf{Z}\|_F^2}{2}\right)^2$, *then we have the following*

$$\nabla h_1(\mathbf{U}, \mathbf{Z}) = \left(\left(\|\mathbf{U}\|_F^2 + \|\mathbf{Z}\|_F^2\right)\mathbf{U}, \left(\|\mathbf{U}\|_F^2 + \|\mathbf{Z}\|_F^2\right)\mathbf{Z}\right) ,$$

$$\langle(\mathbf{H_1}, \mathbf{H_2}), \nabla^2 h_1(\mathbf{U}, \mathbf{Z})(\mathbf{H_1}, \mathbf{H_2})\rangle = (\|\mathbf{H_1}\|_F^2 + \|\mathbf{H_2}\|_F^2)(\|\mathbf{U}\|_F^2 + \|\mathbf{Z}\|_F^2) + 2\left\|\mathbf{H_1 U}^T + \mathbf{ZH_2^T}\right\|_F^2$$

*Proof.* By the definition of Forbenius dot product, we have

$$\frac{1}{4}\|\mathbf{U}\|_F^4 + \frac{1}{4}\|\mathbf{Z}\|_F^4 + \frac{1}{2}\|\mathbf{U}\|_F^2\|\mathbf{Z}\|_F^2 = \frac{1}{4}\langle\mathbf{U}, \mathbf{U}\rangle^2 + \frac{1}{4}\langle\mathbf{Z}, \mathbf{Z}\rangle^2 + \frac{1}{2}\langle\mathbf{U}, \mathbf{U}\rangle\langle\mathbf{Z}, \mathbf{Z}\rangle$$

Now, considering $h_1(\mathbf{U} + \mathbf{H_1}, \mathbf{Z} + \mathbf{H_2})$ we have

$$\frac{1}{4}\langle\mathbf{U} + \mathbf{H_1}, \mathbf{U} + \mathbf{H_1}\rangle^2 + \frac{1}{4}\langle\mathbf{Z} + \mathbf{H_2}, \mathbf{Z} + \mathbf{H_2}\rangle^2 + \frac{1}{2}\langle\mathbf{U} + \mathbf{H_1}, \mathbf{U} + \mathbf{H_1}\rangle\langle\mathbf{Z} + \mathbf{H_2}, \mathbf{Z} + \mathbf{H_2}\rangle$$

$$= \frac{1}{4}\left(\langle\mathbf{U}, \mathbf{U}\rangle + 2\langle\mathbf{H_1}, \mathbf{U}\rangle + \langle\mathbf{H_1}, \mathbf{H_1}\rangle\right)^2 + \frac{1}{4}\left(\langle\mathbf{Z}, \mathbf{Z}\rangle + 2\langle\mathbf{Z}, \mathbf{H_2}\rangle + \langle\mathbf{H_2}, \mathbf{H_2}\rangle\right)^2$$

$$+ \frac{1}{2}\left(\langle\mathbf{U}, \mathbf{U}\rangle + 2\langle\mathbf{H_1}, \mathbf{U}\rangle + \langle\mathbf{H_1}, \mathbf{H_1}\rangle\right)\left(\langle\mathbf{Z}, \mathbf{Z}\rangle + 2\langle\mathbf{Z}, \mathbf{H_2}\rangle + \langle\mathbf{H_2}, \mathbf{H_2}\rangle\right)$$

$$= \frac{1}{4}\left(\langle\mathbf{U}, \mathbf{U}\rangle^2 + 4\langle\mathbf{H_1}, \mathbf{U}\rangle^2 + \langle\mathbf{H_1}, \mathbf{H_1}\rangle^2 + 2\langle\mathbf{H_1}, \mathbf{H_1}\rangle\langle\mathbf{U}, \mathbf{U}\rangle\right.$$

$$\left. + 4\langle\mathbf{U}, \mathbf{U}\rangle\langle\mathbf{H_1}, \mathbf{U}\rangle + 4\langle\mathbf{H_1}, \mathbf{U}\rangle\langle\mathbf{H_1}, \mathbf{H_1}\rangle\right)$$

$$+ \frac{1}{4}\left(\langle\mathbf{Z}, \mathbf{Z}\rangle^2 + 4\langle\mathbf{Z}, \mathbf{H_2}\rangle^2 + \langle\mathbf{H_2}, \mathbf{H_2}\rangle^2 + 2\langle\mathbf{H_2}, \mathbf{H_2}\rangle\langle\mathbf{Z}, \mathbf{Z}\rangle\right.$$

$$\left. + 4\langle\mathbf{Z}, \mathbf{H_2}\rangle\langle\mathbf{Z}, \mathbf{Z}\rangle + 4\langle\mathbf{Z}, \mathbf{H_2}\rangle\langle\mathbf{H_2}, \mathbf{H_2}\rangle\right)$$

$$+ \frac{1}{2}\left(\langle\mathbf{U}, \mathbf{U}\rangle\langle\mathbf{Z}, \mathbf{Z}\rangle + 2\langle\mathbf{U}, \mathbf{U}\rangle\langle\mathbf{Z}, \mathbf{H_2}\rangle + \langle\mathbf{U}, \mathbf{U}\rangle\langle\mathbf{H_2}, \mathbf{H_2}\rangle\right)$$

$$+ \frac{1}{2}\left(2\langle\mathbf{H_1}, \mathbf{U}\rangle\langle\mathbf{Z}, \mathbf{Z}\rangle + 4\langle\mathbf{H_1}, \mathbf{U}\rangle\langle\mathbf{Z}, \mathbf{H_2}\rangle + 2\langle\mathbf{H_1}, \mathbf{U}\rangle\langle\mathbf{H_2}, \mathbf{H_2}\rangle\right)$$

$$+ \frac{1}{2}\left(\langle\mathbf{H_1}, \mathbf{H_1}\rangle\langle\mathbf{Z}, \mathbf{Z}\rangle + 2\langle\mathbf{H_1}, \mathbf{H_1}\rangle\langle\mathbf{Z}, \mathbf{H_2}\rangle + \langle\mathbf{H_1}, \mathbf{H_1}\rangle\langle\mathbf{H_2}, \mathbf{H_2}\rangle\right)$$

Collecting all the first order terms, we have

$$\langle\mathbf{U}, \mathbf{U}\rangle\langle\mathbf{H_1}, \mathbf{U}\rangle + \langle\mathbf{Z}, \mathbf{H_2}\rangle\langle\mathbf{Z}, \mathbf{Z}\rangle + \langle\mathbf{U}, \mathbf{U}\rangle\langle\mathbf{Z}, \mathbf{H_2}\rangle + \langle\mathbf{H_1}, \mathbf{U}\rangle\langle\mathbf{Z}, \mathbf{Z}\rangle ,$$

and similarly collecting all the second order terms we have

$$\frac{1}{4}\left(4\langle\mathbf{H_1}, \mathbf{U}\rangle^2 + 2\langle\mathbf{H_1}, \mathbf{H_1}\rangle\langle\mathbf{U}, \mathbf{U}\rangle + 4\langle\mathbf{Z}, \mathbf{H_2}\rangle^2 + 2\langle\mathbf{H_2}, \mathbf{H_2}\rangle\langle\mathbf{Z}, \mathbf{Z}\rangle\right)$$

$$+ \frac{1}{2}\left(\langle\mathbf{U}, \mathbf{U}\rangle\langle\mathbf{H_2}, \mathbf{H_2}\rangle + 4\langle\mathbf{H_1}, \mathbf{U}\rangle\langle\mathbf{Z}, \mathbf{H_2}\rangle + \langle\mathbf{H_1}, \mathbf{H_1}\rangle\langle\mathbf{Z}, \mathbf{Z}\rangle\right) ,$$

$$= \frac{1}{2}\left(2\langle\mathbf{H_1}, \mathbf{U}\rangle^2 + (\langle\mathbf{H_1}, \mathbf{H_1}\rangle + \langle\mathbf{H_2}, \mathbf{H_2}\rangle)(\langle\mathbf{U}, \mathbf{U}\rangle + \langle\mathbf{Z}, \mathbf{Z}\rangle)\right.$$

$$\left. + 2\langle\mathbf{Z}, \mathbf{H_2}\rangle^2 + 4\langle\mathbf{H_1}, \mathbf{U}\rangle\langle\mathbf{Z}, \mathbf{H_2}\rangle\right) ,$$

$$= \frac{1}{2}\left((\langle\mathbf{H_1}, \mathbf{H_1}\rangle + \langle\mathbf{H_2}, \mathbf{H_2}\rangle)(\langle\mathbf{U}, \mathbf{U}\rangle + \langle\mathbf{Z}, \mathbf{Z}\rangle) + 2(\langle\mathbf{H_1}, \mathbf{U}\rangle + \langle\mathbf{Z}, \mathbf{H_2}\rangle)^2\right) .$$

Thus the statement follows. $\qquad\square$

**Lemma G.3.** *Given* $h_2(\mathbf{U}, \mathbf{Z}) := \frac{\|\mathbf{U}\|_F^2 + \|\mathbf{Z}\|_F^2}{2}$, *then we have the following*

$$\nabla h_2(\mathbf{U}, \mathbf{Z}) = (\mathbf{U}, \mathbf{Z}),$$

$$\langle (\mathbf{H_1}, \mathbf{H_2}), \nabla^2 h_2(\mathbf{U}, \mathbf{Z})(\mathbf{H_1}, \mathbf{H_2}) \rangle = \|\mathbf{H_1}\|_F^2 + \|\mathbf{H_2}\|_F^2 .$$

*Proof.* Considering $h_2(\mathbf{U} + \mathbf{H_1}, \mathbf{Z} + \mathbf{H_2})$, we have

$$\frac{1}{2} \langle \mathbf{U} + \mathbf{H_1}, \mathbf{U} + \mathbf{H_1} \rangle + \frac{1}{2} \langle \mathbf{Z} + \mathbf{H_2}, \mathbf{Z} + \mathbf{H_2} \rangle$$
$$= \frac{1}{2} \left( \langle \mathbf{U}, \mathbf{U} \rangle + 2 \langle \mathbf{U}, \mathbf{H_1} \rangle + \langle \mathbf{H_1}, \mathbf{H_1} \rangle \right) + \frac{1}{2} \left( \langle \mathbf{Z}, \mathbf{Z} \rangle + 2 \langle \mathbf{Z}, \mathbf{H_2} \rangle + \langle \mathbf{H_2}, \mathbf{H_2} \rangle \right) .$$

Collecting all the first order terms we have

$$\langle \mathbf{U}, \mathbf{H_1} \rangle + \langle \mathbf{Z}, \mathbf{H_2} \rangle ,$$

and similarly collecting all the second order terms we have

$$\frac{1}{2} \left( \langle \mathbf{H_1}, \mathbf{H_1} \rangle + \langle \mathbf{H_2}, \mathbf{H_2} \rangle \right) .$$

Thus the statement holds. $\qquad\square$

## G.1 Proof of Proposition 2.1

*Proof.* We prove here the convexity of $Lh_a - g$ for a certain constant $L \geq 1$. With Lemma G.1 we obtain

$$\langle (\mathbf{H}_1, \mathbf{H}_2), \nabla^2 g(\mathbf{A}, \mathbf{UZ})(\mathbf{H}_1, \mathbf{H}_2) \rangle$$
$$= \|\mathbf{H}_1 \mathbf{Z} + \mathbf{U} \mathbf{H}_2\|_F^2 - 2 \langle \mathbf{A} - \mathbf{UZ}, \mathbf{H}_1 \mathbf{H}_2 \rangle ,$$
$$\leq 2 \|\mathbf{H}_1 \mathbf{Z}\|_F^2 + 2 \|\mathbf{U} \mathbf{H}_2\|_F^2 + 2 \|\mathbf{A}\|_F \|\mathbf{H}_1 \mathbf{H}_2\|_F + 2 \|\mathbf{UZ}\|_F \|\mathbf{H}_1 \mathbf{H}_2\|_F ,$$
$$\leq 2 \|\mathbf{H}_1\|_F^2 \|\mathbf{Z}\|_F^2 + 2 \|\mathbf{U}\|_F^2 \|\mathbf{H}_2\|_F^2 + 2 \|\mathbf{A}\|_F \|\mathbf{H}_1\|_F \|\mathbf{H}_2\|_F + 2 \|\mathbf{U}\|_F \|\mathbf{Z}\|_F \|\mathbf{H}_1\|_F \|\mathbf{H}_2\|_F .$$

With AM-GM inequality, for non-negative real numbers $a, b$ we have $2\sqrt{ab} \leq a + b$, we have

$$2 \|\mathbf{U}\|_F \|\mathbf{Z}\|_F \|\mathbf{H}_1\|_F \|\mathbf{H}_2\|_F \leq \|\mathbf{H}_1\|_F^2 \|\mathbf{Z}\|_F^2 + \|\mathbf{U}\|_F^2 \|\mathbf{H}_2\|_F^2 ,$$

and similarly we have

$$2 \|\mathbf{A}\|_F \|\mathbf{H}_1\|_F \|\mathbf{H}_2\|_F \leq \|\mathbf{A}\|_F \|\mathbf{H}_1\|_F^2 + \|\mathbf{A}\|_F \|\mathbf{H}_2\|_F^2 .$$

Using the above two inequalities, we obtain

$$\langle (\mathbf{H}_1, \mathbf{H}_2), \nabla^2 g(A, \mathbf{UZ})(\mathbf{H}_1, \mathbf{H}_2) \rangle \leq (3 \|\mathbf{Z}\|_F^2 + \|\mathbf{A}\|_F) \|\mathbf{H}_1\|_F^2 + (3 \|\mathbf{U}\|_F^2 + \|\mathbf{A}\|_F) \|\mathbf{H}_2\|_F^2 .$$
$$\tag{G.1}$$

Now, considering the kernel generating distances, via Lemma G.2 and G.3 we obtain

$$\langle (\mathbf{H}_1, \mathbf{H}_2), \nabla^2 h_1(\mathbf{U}, \mathbf{Z})(\mathbf{H}_1, \mathbf{H}_2) \rangle$$
$$= 2 \|\mathbf{H}_1 \mathbf{U} + \mathbf{H}_2 \mathbf{Z}\|_F^2 + (\|\mathbf{U}\|_F^2 + \|\mathbf{Z}\|_F^2) \|\mathbf{H}_1\|_F^2 + (\|\mathbf{U}\|_F^2 + \|\mathbf{Z}\|_F^2) \|\mathbf{H}_2\|_F^2$$
$$\geq \|\mathbf{Z}\|_F^2 \|\mathbf{H}_1\|_F^2 + \|\mathbf{U}\|_F^2 \|\mathbf{H}_2\|_F^2 ,$$

and

$$\langle (\mathbf{H}_1, \mathbf{H}_2), \nabla^2 h_2(\mathbf{U}, \mathbf{Z})(\mathbf{H}_1, \mathbf{H}_2) \rangle = \|\mathbf{H}_1\|_F^2 + \|\mathbf{H}_2\|_F^2 .$$

Now, it is easy to see that

$$\langle (\mathbf{H}_1, \mathbf{H}_2), \nabla^2 h_a(\mathbf{U}, \mathbf{Z})(\mathbf{H}_1, \mathbf{H}_2) \rangle \geq \langle (\mathbf{H}_1, \mathbf{H}_2), \nabla^2 g(\mathbf{A}, \mathbf{UZ})(\mathbf{H}_1, \mathbf{H}_2) \rangle .$$

A similar proof holds for the convexity of $Lh_a + g$, however the choice of $L$ here need not be the same as it is for $Lh_a - g$ (see [9, Remark 2.1]). $\qquad\square$

# H    Additional Experiments and Implementation Details

## H.1    Double Backtracking Implementation

This subsection where we provide certain crucial implementation details of CoCaIn BPG-MF algorithm, is largely based on [46, Section 5.4]. Note that CoCaIn BPG-MF is a sequential algorithm in the sense one can compute $Y_{\mathbf{U}}^{\mathbf{k}}, Y_{\mathbf{Z}}^{\mathbf{k}}$ first via the steps (2.8), (2.9) and (2.10). Then, the updates can be done exactly like BPG-MF, where step-size depends on the parameter $\bar{L}_k$ obtained via (2.12). In (2.10) it is required to find $\underline{L}_k$ such that the following holds

$$D_g\left(\mathbf{U^{k+1}}, \mathbf{Z^{k+1}}, Y_{\mathbf{U}}^{\mathbf{k}}, Y_{\mathbf{Z}}^{\mathbf{k}}\right) \geq -\underline{L}_k D_h\left(\mathbf{U^{k+1}}, \mathbf{Z^{k+1}}, Y_{\mathbf{U}}^{\mathbf{k}}, Y_{\mathbf{Z}}^{\mathbf{k}}\right), \tag{H.1}$$

similarly in (2.12) it is required to find $\bar{L}_k$ such that

$$D_g\left(\mathbf{U^{k+1}}, \mathbf{Z^{k+1}}, Y_{\mathbf{U}}^{\mathbf{k}}, Y_{\mathbf{Z}}^{\mathbf{k}}\right) \leq \bar{L}_k D_h\left(\mathbf{U^{k+1}}, \mathbf{Z^{k+1}}, Y_{\mathbf{U}}^{\mathbf{k}}, Y_{\mathbf{Z}}^{\mathbf{k}}\right). \tag{H.2}$$

The above mentioned steps can be solved via the classical backtracking strategy for $\underline{L}_k$ and $\bar{L}_k$ individually, hence the name "double backtracking". We describe the backtracking procedure for $\underline{L}_k$ and it is easy to extend to $\bar{L}_k$. The backtracking strategy involves a scaling parameter $\nu \geq 1$ and an initialization point $\underline{L}_{k,0} > 0$ (preferably small) both chosen by the user and the parameter $\underline{L}_k$ is set to the smallest element from the set $\left\{\underline{L}_{k,0}, \nu\underline{L}_{k,0}, \nu^2\underline{L}_{k,0}, \dots\right\}$ such that (2.10) holds. For $\bar{L}_k$ one requires to use (2.12) and also due to the additional restriction that $\bar{L}_k \geq \bar{L}_{k-1}$ in CoCaIn BPG-MF it is required to start the initialization $\bar{L}_{k,0} = \bar{L}_{k-1}$.

## H.2    Non-negative Matrix Factorization

We consider the same setting as the simple matrix factorization problem considered in 3, however we set $\mathcal{U} = \mathbb{R}_+^{M \times K}$ and $\mathcal{Z} = \mathbb{R}_+^{K \times N}$. We consider Medulloblastoma dataset [12] dataset with matrix $A \in \mathbb{R}^{5893 \times 34}$. As evident from Figure 4 PALM based methods outpeform BPG methods here. This raises new open questions and hints at potential variants of BPG for constrained problems with global convergence.

| (a) No-Regularization | (b) L2-Regularization | (c) L1-Regularization |

Figure 4: **Non-negative Matrix Factorization on Medulloblastoma Dataset [12].**

## H.3    Matrix Completion

The MovieLens datasets are essentially a matrix $A \in \mathbb{R}^{M \times N}$, where $M$ denotes the number of users and $N$ denotes the number of movies. Only a few non-zero entries are given and the entries denote the ratings which the user has provided for a particular movie. The ratings can take the value between 1 and 5, which we refer to as scale. The exact statistics of all the MovieLens datasets are given below.

| Dataset | Users | Movies | Non-zero entries | Scale |
|---|---|---|---|---|
| MovieLens100K | 943 | 1682 | 100000 | 1-5 |
| MovieLens1M | 6040 | 3952 | 1000209 | 1-5 |
| MovieLens10M | 71567 | 10681 | 10000054 | 1-5 |

The plots provided for the matrix completion problem in Section 3 uses only 80% of the data and we use the remaining 20% as test data in order to obtain the generalization performance to unseen matrix

entries with the resulting factors $\mathbf{U} \in \mathbb{R}^{M \times K}$ and $\mathbf{Z} \in \mathbb{R}^{K \times N}$ where we use $K = 5$. The predicted rating to a particular $i \in \{1, 2, \ldots, M\}$ and $j \in \{1, 2, \ldots, N\}$ is given by $(\mathbf{UZ})_{ij}$. The test data is comprised of matrix indices with unseen entries and we denote this set of indices as $\Omega_T$. A popular measure for the test data is the Test RMSE, which is given by the following entity

$$\text{Test RMSE} = \sqrt{\frac{1}{|\Omega_T|} \sum_{i=1}^{M} \sum_{j=1}^{N} \mathbf{I}_{(i,j) \in \Omega_T} \left( \mathbf{A}_{ij} - (\mathbf{UZ})_{ij} \right)^2}$$

where $|\Omega_T|$ denotes the cardinality of the set $\Omega_T$ and $\mathbf{I}_{(i,j) \in \Omega_T} = 1$ if the index pair $(i, j)$ lies in the set $\Omega_T$ else it is zero. The Test RMSE comparisons for the MovieLens Dataset are given below in Figure 5.

| (a) MovieLens-100K | (b) MovieLens-1M | (c) MovieLens-10M |

Figure 5: **Test RMSE plot on MovieLens Datasets [30].**

The above given figures show that the proposed methods BPG-MF-WB and CoCaIn BPG-MF are competitive to PALM and iPALM. BPG-MF is slow in the beginning, however it is competitive to other methods towards the end.

## H.4 Time Comparisons

We provide time comparisons in Figures 6, 7, 8 for all the experimental settings mentioned in Section 3, where we mention the dataset in the caption. Since, we used logarithmic scaling, we used an offset of $10^{-2}$ for all algorithms for better visualization.

| (a) No Regularization | (b) L2-Regularization | (c) L1-Regularization |

Figure 6: **Time plots for Simple Matrix Factorization on Synthetic Dataset.**

As evident from the plots, the proposed variants BPG-MF-WB and CoCaIn BPG-MF are competitive that PALM and iPALM. And, BPG-MF is mostly slow, due to constant step-size, which can be potentially helpful when backtracking is computationally expensive.

(a) No-Regularization      (b) L2-Regularization      (c) L1-Regularization

Figure 7: **Time plots for Non-negative Matrix Factorization on Medulloblastoma dataset [12].**

(a) MovieLens-100K      (b) MovieLens-1M      (c) MovieLens-10M

Figure 8: **Time plots for Matrix Completion on MovieLens Datasets [30].**