[Reviews · NeurIPS 2019]

Reviewer 1



I've read the response and the justification for assumptions A and B are nice. The score has increased to 7. ---------------------------------------------------------------------- 1. Originality: This paper devises a new distance function applied in BGP algorithm. The framework is existing but the author gives a number of improvements and for the first time gives convergence theorem under some conditions. 2. Quality: The theoretical derivation looks sound and the experiments are rather comprehensive. 3. Clarity: The text is mainly clear, but the statement of BPG-MF algorithm is a little confusing --- one needs to read through many sections to get the whole algorithm. 4. Significance: The results are theoretically valuable as a non-alternating optimization method with thorough analysis. But in real data experiments this algorithm does not show a significant advantage in performance over the traditional methods, especially when computing resource is limited.

Reviewer 2



In this study, inertial Bregman proximal gradient algorithms (BPG and CoCaIn-BPG) were considered for matrix factorization. The problem here was the existence of L-smad property in matrix factorization. To tackle this issue, Authors proposed a novel Bregman distance, and proved the L-smad property with this distance and infer convergence of BPG and CoCaIn BPG. I think the contributions in this paper is novel and significant. In experiments, three problems (matrix factorization, non-negative matrix factorization, and matrix completion) were solved by the proposed optimization algorithms and the effectiveness compared with state-of-the-art algorithms such as PALM, iPALM was demonstrated. I think it was nice if the convergence behaviors are statistically evaluated because the stationary points obtained by non-convex optimization are usually varied by initialization. Thus, I claim that the experiments are slightly weak.

Reviewer 3



First, some background. The Bregman Proximal Gradient (BPG) method was introduced in general form with theory by Bolte et al. in [8] for nonconvex optimization, and an inertial version with convex-concave backtracking (CoCaInBPG) was introduced by Mukkamala et al. in [45]. In section 6.6 of the same paper, Mukkamala et al. apply CoCaInBPG to the problem of structured matrix factorization and show good performance, but leave the theory and computational efficiency open. Additionally, BPG without inertia was applied sucessfully to the problem of symmetric non-negative matrix factorization (SNMF) by Dragomir et al. in [20], which appears in its most recent version to also include more general (symmetric) matrix completion. The submitted paper here is concerned with applying BPG and CoCaInBPG to the non-symmetric matrix factorization problem, essentially picking up where [45] left off and providing work complementary to [20]. This paper first restates the results of [8,45] for the specific objective (1.1) of matrix factorization, then makes its two primary contributions. First, the paper introduces a kernel generating distance function h that is appropriate for matrix factorization, which is related to the universal function of [20] but extended non-trivially to the non-symmetric case. This proves convergence theory to a stationary point of the method with this distance function. Second, the paper introduces closed form solutions for the updates corresponding to a variety of different variants of (1.1) with this kernel. This makes the methods usable in practice and makes the theory more interesting and is a welcome contribution. The quality and clarity of the paper is in general high. It is well-written (barring the need for a once-over to fix a few typos) and easy to understand, and the results are well-presented. This is not a critique of the paper itself, but I would like to note that there is something wrong with the citation system that made this paper slightly arduous to review. For example, on L29 it appears [4] is cited twice in one citation, but in fact on further inspection it appears that half the time that [4] is cited the appropriate paper is actually [45] -- in other words, CoCaIn BPG was introduced by Mukkamala et al. (https://arxiv.org/abs/1904.03537) and not by Bauschke et al. (Mathematics of Operations Research, 42(2):330–348, 2017.). In fact, every citation is truncated to the first digit of the citation number. It is possible that this is a rendering error on my machine, but I wanted to bring this to the authors' attention. Additionally, as a general style comment the citations appear copious in general when it comes to repeating citations for the same paper at multiple places in a paragraph. I did not check all the math in the supplementary material, but I trust that the results the authors give are correct. The form of the Bregman generating function in section 2.2 seems to make sense in light of the relation to the universal kernel of [20], and the closed-form updates as well seem morally correct. I do wish that more time was spent discussing the overall time-complexity of the algorithm beyond what is mentioned in line 241 -- in particular, adding an explicit line regarding discussion of whether the cost of an iteration is really comparable between methods or not would be great (and the timing results in the appendix seem promising!). The results seem original, as there is clearly recent and parallel related work but the non-symmetric case treated here is more general in some ways. What I am not entirely sure about is the significance of the results here. In particular, it is difficult to understand what point the numerical results in section 3 are trying to make ("We aim to empirically illustrate the differences of alternating minimization 257 based schemes and non-alternating schemes[.]"). Very many of these results seem to show competing methods converging to a different stationary point than the methods of this paper. However, that is the nature of non-convex optimization. I do not believe that the authors have theory to say that the non-alternating schemes should converge to a "better" stationary point, so it doesn't really seem like we can draw too much of a conclusion from Figure 1 or Figure 2. In other words, why should we choose the schemes of this paper over other schemes? Is there any reason we should expect the results of these tests to generalize? What makes BPG / CoCaIn BPG a better choice of optimization algorithm than PALM-methods in general? However, not every method has to be immediately "SOTA" and so, that said, I think the paper is a nice contribution. -- Having read the authors' response and the other reviewers comments, I am quite happy with the direction of the final paper. The authors clearly outlined a number of substantial benefits of their approach over competing methods and promised to add these to an extended discussion section. Further, they have put together an additional numerical experiment to demonstrate more clearly the improved empirical performance over competing methods, and the initial results they report are promising. I am raising my score, as the authors have addressed my primary points from (5) below.

[Author Response · NeurIPS 2019]

We sincerely appreciate the effort given by reviewers for the time to read, review and provide insightful comments. We will fix all the typos (in particular Eq.(2.5)). We thank the reviewers for pointing out the crucial bug in the display of citations. In general, we will improve the presentation, the experimental section and provide open source code. We now address the reviewers questions in detail.

**Assumptions A and B are mostly satisfied in practice.** Assumption A where $C$ is the full space and $h = \frac{1}{2}\|\cdot\|^2$ is the standard setting of Proximal Gradient Descent. Examples such as QIP [8] and Cubic Regularization [45] satisfy the assumption for non-Euclidean function $h$. For the generalization to strict subsets $C$ of $\mathbb{R}^d$, Assumption A uses a flexible formulation that does not require the function $g$ to be smooth on the full space (only required on an open set containing $\mathrm{dom}\,h$). Also $\mathrm{dom}\,f \cap C$ has to be non-empty otherwise the optimization will take place over an empty set, which is not desirable. Moreover, one is often interested in reaching global minimum and do not expect to decrease the function value to negative infinity, hence the lower-boundedness. Assumption B ensures that the updates of BPG-methods are well-defined with mild requirements as explained on page 6 in [8], for example, under a classical constraint qualification.

**Statistical evaluation experiments.** From the empirical results given in the paper, BPG-methods are seemingly better than (or at least as good as) PALM-methods. We believe that the trajectories taken by both the methods are different and is source of good performance of BPG-methods over PALM-methods. The theoretical justification however is an open question, which we could not justify. We agree that statistical evaluation varied with initialization is an important experiment and we will add the proposed experiments. With these additional experiments, we hope to find some new insights (similar to preliminary results below) on the convergence behavior of BPG-methods vs PALM-methods. Our new preliminary statistical evaluation results with setting used in "Figure 2 with No regularization" show that PALM-methods get stuck at large objective values ($\approx$ 96728.941 on average) however BPG-methods do not ($\approx$ 378.173 on average). We will give detailed experiments along with precise settings in the next update.

**Computational benefits and insights.** We briefly remark some properties of the update steps of BPG-methods. Note that the updates are independent for $\mathbf{U}$ and $\mathbf{Z}$, for example, see the simple illustration on Page 2, where updates can be done in parallel blockwise (ignoring the 1D cubic equation). This should increase the speedup in practice, in particular for large matrices. Also, note that some terms in gradients overlap, so using temporary variables in implementation can increase the speedup. We will discuss these tricks, the time and computational complexity in detail in conjunction with the timing experiments along with a brief pointer in the main paper, as per suggestion by Reviewer 3. We now provide insights on why BPG-methods are a better choice over other methods, with focus on alternating methods.

- PALM-methods estimate a Lipschitz constant with respect to a block of coordinates in each iteration, which is expensive for large block matrices. BPG-methods use a global L-smad constant, which is efficient.

- PALM-methods cannot be parallelized block wise, for example, in the two block case, the computation of the Lipschitz constant of second block must wait for the first block to be updated, hence it is inherently serial.

- Alternating minimization methods do not converge for non-smooth regularization terms and can be inefficient (for, e.g., ALS) for some matrix factorization problems (see, for example, "Tensor Decompositions and Applications" by T. G. Kolda and B. W. Bader, also see "On search directions for minimization algorithms" by M.J.D. Powell). But, BPG-methods and PALM-methods converge (due to linearization).

- PALM is not applicable to the 2D function $g(x, y) = (x^3 + y^3)^2$, because the block-wise Lipschitz continuity of the gradients fails to hold even after fixing one variable. BPG-methods are applicable here.

- PALM is not applicable to, for example, symmetric Matrix Factorization as also pointed in [20] or the following penalty method based (relaxed) orthogonal NMF problem (similar to Eq.1.1)

$$\min_{\mathbf{U} \in \mathcal{U}, \mathbf{Z} \in \mathcal{Z}} \left\{ \Psi \equiv \frac{1}{2} \|\mathbf{A} - \mathbf{UZ}\|_F^2 + \frac{\rho}{2} \left\|\mathbf{U}^T\mathbf{U} - \mathbf{I}\right\|_F^2 + \mathbf{I}_{\mathbf{U} \geq \mathbf{0}} + \mathbf{I}_{\mathbf{Z} \geq \mathbf{0}} + \mathcal{R}_1(\mathbf{U}) + \mathcal{R}_2(\mathbf{Z}) \right\}, \quad (1)$$

  where second term does not have a block-wise Lipschitz continuous gradient for any $\rho > 0$. But, here BPG-methods are applicable (similarly also for Projective NMF) with minor changes to the Bregman distance.

- The block wise Lipschitz continuity of gradients was the primary motive for PALM-methods. Now, with the BPG-methods, we can directly tackle the original problem with L-smad property.

- BPG-methods are very general so the choice of applications will increase substantially and we believe that this will open doors to design new losses and regularizers, without restricting to Lipschitz continuous gradients.

We agree that an expanded discussion of competing methods is crucial for the paper. We will also discuss the points mentioned in the previous answers in detail in the expanded discussion along with state of the art matrix factorization models and new perspectives with BPG-methods.

[Meta-Review · NeurIPS 2019]

The paper introduces a new function generating a Bregman distance and develop Bregman proximal gradient methods applied to various matrix factorization problems. The authors clearly outline the benefits of the approach and computationally demonstrate this.